

Highly time-resolved urban aerosol characteristics during
springtime in Yangtze River Delta, China: Insights from soot
particle aerosol mass spectrometry
Junfeng Wang,[1] Xinlei Ge,[1,]* Yanfang Chen,[1] Yafei Shen,[1] Qi Zhang,[1,2] Yele Sun,[3]
Jianzhong Xu,[4] Huan Yu,[1] Mindong Chen[1,*]
[1]Jiangsu Key Laboratory of Atmospheric Environment Monitoring and Pollution
Control (AEMPC), Collaborative Innovation Center of Atmospheric Environment and
Equipment Technology (CIC-AEET), School of Environmental Science and
Engineering, Nanjing University of Information Science & Technology, Nanjing
210044, China
[2]Department of Environmental Toxicology, University of California at Davis, Davis,
California 95616, United States
[3]State Key Laboratory of Atmospheric Boundary Layer Physics and Atmospheric
Chemistry, Institute of Atmospheric Physics, Chinese Academy of Sciences, Beijing
100029, China
[4]State Key Laboratory of Cryospheric Sciences, Cold and Arid Regions,
Environmental and Engineering Research Institute, Chinese Academy of Sciences,
Lanzhou 730000, China
*Corresponding author, Email: caxinra@163.com; chenmdnuist@163.com

23              Phone: +86-25-58731394

For *Atmos. Chem. Phys.*



**Abstract:** In this work, the Aerodyne soot particle – aerosol mass spectrometer
(SP-AMS) was deployed for the first time during the spring of 2015 in urban Nanjing,
a megacity in the Yangtze River Delta (YRD) of China, for online characterization of
the submicron aerosols ($PM_1$). The SP-AMS enables real-time and fast quantification
of refractory black carbon ($r$BC) simultaneously with other non-refractory species
(ammonium, sulfate, nitrate, chloride and organics). The average $PM_1$ concentration
was found to be 28.2 μg m$^{-3}$ (~54% of the $PM_{2.5}$ mass), with organics (45%) as the
most abundant component, following by sulfate (19.3%), nitrate (13.6%), ammonium
(11.1%), $r$BC (9.7%) and chloride (1.3%). These $PM_1$ species together can reconstruct
~44% of the light extinction during this campaign based on the IMPROVE method.
Chemically-resolved mass-based size distributions revealed that small particles
especially ultrafine ones (<100 nm vacuum aerodynamic diameter) were dominated
by organics and $r$BC, while large particles had significant contributions from
secondary inorganic species. Source apportionment of organic aerosols (OA) yielded
four OA subcomponents, including hydrocarbon-like OA (HOA), cooking-related OA
(COA), semi-volatile oxygenated OA (SV-OOA), and low-volatility oxygenated OA
(LV-OOA). Overall, secondary organic aerosol (SOA, equal to the sum of SV-OOA
and LV-OOA) dominated the total OA mass (55.5%), but primary organic aerosol
(POA, equal to the sum of HOA and COA) can outweigh SOA in early morning and
evening due to enhanced human activities. High OA concentrations were often
associated with high mass fractions of POA and $r$BC, indicating the important role of
anthropogenic emissions during heavy pollution events. The diurnal cycles of nitrate,
chloride and SV-OOA both showed good anti-correlations with air temperatures,
suggesting their variations were likely driven by thermodynamic equilibria and
gas-to-particle partitioning. On the other hand, in contrast to other species, sulfate and
LV-OOA concentrations increased during afternoon, and showed no positive
correlations with relative humidity (RH), indicating the significant role of
photochemical processing rather than aqueous-phase processing for their formations.
The bivariate polar plots show that the SV-OOA was formed locally, and the



variations of hydrogen-to-carbon (H/C) and oxygen-to-carbon (O/C) ratios in the Van
Krevelen space further suggests an evolution pathway of SV-OOA to LV-OOA. Our
findings regarding springtime aerosol chemistry in Nanjing may have important
implications for the air quality remediation in the densely populated regions.

**1. Introduction**

In recent years, high concentrations of fine particulate matter ($PM_{2.5}$) have been

frequently observed (Hu et al., 2015), in accompanying with the visibility impairment
and occurrence of haze events across large parts of China. $PM_{2.5}$ also affects human
health (e.g., Pope and Dockery, 2006;Cao et al., 2012), regional and global climate
(directly by absorbing and scattering solar radiation or indirectly by acting as cloud
condensation nuclei and ice nuclei)(e.g.,Ghan and Schwartz, 2007;Pöschl, 2005), and
the earth's ecosystem (Carslaw et al., 2010). These effects are predominantly
dependent upon the physical and chemical characteristics of fine particles, such as
mass concentration, chemical composition, size distribution, and hygroscopicity, all of
which are influenced by the emission sources and transformation and evolution
processes in the atmosphere.

The Yangtze River Delta (YRD) region is one of the most populated and

economically developed areas in China, but it is also facing with severe air pollution
lately. Nanjing, as one of the major megacities in this region, has a daily $PM_{2.5}$ mass
concentration varying between 33-234 $\mu g\ m^{-3}$ during November 2011 - August 2012,
with an mean value of 106 $\mu g\ m^{-3}$, which is 4.2 times the WHO air quality standard of
25 $\mu g\ m^{-3}$ (Shen et al., 2014). $PM_{2.5}$ pollution is significantly elevated during hazy
days, for example, a daily average of 282 $\mu g\ m^{-3}$ was observed for a heavily polluted
day (Fu et al., 2008). A number of studies regarding aerosol chemistry in Nanjing
have been conducted, and identified various inorganic components (sulfate, nitrate,
ammonium and heavy metals, etc.) (e.g., Wang et al., 2003;Hu et al., 2012) and
hundreds of organic species (carboxylic/dicarboxylic acids, amines and amino acids,
polycyclic aromatic hydrocarbons, etc.) (Wang et al., 2011;Wang et al., 2002;Yang et



al., 2005;Wang et al., 2009) that contribute to the aerosol mass. However, past studies
mostly employed filter-based sampling technique, which due to low time resolution (a
few hours to days), is often incapable of capturing details of the atmospheric
evolution processes during the typical lifecycle of aerosols (Wexler and Johnston,
2008). Subsequent offline analyses may also introduce artifacts as some semi-volatile
species can evaporate during sampling and storage (Dong et al., 2012).

On the other hand, in the past 15 years, the Aerodyne Aerosol Mass spectrometer

(AMS) (Canagaratna et al., 2007) has been widely used, and was proven to be
powerful for real-time online measurements of size-resolved chemical compositions
of submicron aerosols (PM$_1$) with very fine time resolution (seconds to minutes)
(Zhang et al., 2007a;Jimenez et al., 2009). The development of Aerodyne AMS began
with the invention of quadruple AMS (Q-AMS) (Jayne et al., 2000), following by the
compact time-of-flight AMS (C-ToF-AMS) (Drewnick et al., 2005), high resolution
time-of-flight AMS (HR-ToF-AMS) (DeCarlo et al., 2006) and the soot particle AMS
(SP-AMS) (Onasch et al., 2012). There are also an aerosol chemical speciation
monitor (ACSM) (Ng et al., 2011) and its updated version of ToF-ACSM (Fröhlich et
al., 2013), which are in particular designed for long-term unattended aerosol
measurements. SP-AMS is the most advanced version, which in principle incorporates
the single particle soot photometer (SP2) into the HR-ToF-AMS, and upgraded with a
laser vaporizer for detecting refractory black carbon ($r$BC) and associated/coated
species that cannot be measured by other types of AMS.

Recently, the Aerodyne AMS has been deployed widely in China (particularly

Beijing) (e.g., Xu et al., 2014 and references therein;Sun et al., 2014;Yeung et al.,
2014;Zhang et al., 2014;Li et al., 2015;Shen et al., 2015;Sun et al., 2015a;Sun et al.,
2015b;Yan et al., 2015;Zhang et al., 2015;Tang et al., 2016;Zhang et al., 2016a;Jiang
et al., 2015;Chen et al., 2015;Xu et al., 2015;Du et al., 2015;Sun et al., 2016;Wang et
al., 2015;Han et al., 2015;Wang et al., 2016b). However, only a few field campaigns
were conducted in the YRD region. Huang et al. (2012b) deployed an HR-ToF-AMS
together with an SP2 in Shanghai during the 2010 Shanghai World Expo, and in



Jiaxing during summer and winter of 2010 (Huang et al., 2012a). In urban Nanjing, an
ACSM was applied for characterizing $PM_1$ during summer and autumn harvest
seasons (Zhang et al., 2015), and during December 2013 to investigate a few heavy
haze events (Zhang et al., 2016b). In addition, a Q-AMS was deployed in Nanjing to
investigate the effects of $PM_1$ on visibility during January 2013 (Shen et al., 2015).
Furthermore, a recent study by Wang et al. (2016a) reported the observation of
fullerene soot in suburban Nanjing using an SP-AMS. Nevertheless, many questions
remain with regard to aerosol chemistry, sources, and processes in this region.
Moreover, none of the previous AMS measurements studied the aerosol
characteristics during springtime in Nanjing. For these reasons, we reports in this
work, for the first time, the real-time measurement results on urban fine aerosols in
Nanjing using the SP-AMS during spring in 2015. The rich high resolution mass
spectra (HRMS) data allow us to conduct in-depth analyses, and better understand the
characteristics, sources and relevant transformation processes of ambient aerosols in
Nanjing.

**2. Experiments**
**2.1 Sampling site and instrumentation**
The field campaign was conducted in the environment monitoring station of
Nanjing Olympic center (32°0′33.00″N, 118°44′9.53″E, Fig. S1) from April 13 to 29,
2015. Details of the sampling site are shown in Fig. S1. The site was surrounded by
residential buildings, close to a few urban arterial roads (~ 85 m northwest of
Huangshan Road, ~ 200 m northeast to Mengdu Street and ~425 m southwest of
Xinglong Street). There are also a restaurant (~50 m), a student cafeteria (~300 m),
and the Nanjing Cigarette Factory (~480 m southeast) around the site.
The sampling inlet was installed outside the fifth floor of the building (~12 m
above the ground), with a $PM_{2.5}$ cyclone (URG Corp., Chapel Hill, NC, USA) to
remove coarse particles. Ambient particles were dried (RH <10%) via a diffusion
dryer filled with silica gel before entering into the SP-AMS. The sampling line (~2 m



long) was assembled using stainless steel tubing and proper fittings. Air flow was
controlled at around ~5 L min$^{-1}$, with a flow rate into the SP-AMS at ~80 cm$^3$ min$^{-1}$.

The SP-AMS can measure non-refractory (NR) PM$_1$ components including

ammonium, nitrate, sulfate, chloride and organics similar to other types of AMS via a
thermal tungsten heater. Moreover, it can also measure $r$BC and coated species as it is
equipped with an intracavity Nd:YAG laser vaporizer (1064 nm) (Onasch et al., 2012).
During this campaign, the instrument was switched between "laser on" and "laser off"
settings, and between V-mode (better for mass quantification) and W-mode (better
chemical resolution, ~5000 in this study), with one cycle including six menu settings
(M1: Laser on V-mode; M2: Laser off V-mode; M3: Laser on W-mode; M4: Laser off
W-mode; M7: Laser on PToF-mode; M8: Laser off PToF-mode). Each menu was set
to 2.5 min, thus a full running cycle lasted for 15 mins. The PToF-mode was under
V-mode, but was tuned in particular for measuring particle sizes. The tungsten heater
was always turned on and kept at ~600$^o$C.

The SP-AMS, in conjunction with a scanning mobility particle sizer (SMPS) (TSI

inc., Shoreview, MN, USA) was calibrated for mass quantification (e.g., ionization
efficiency) using size-selected (250 nm and 300 nm) monodisperse ammonium nitrate
particles following the procedures detailed in Jimenez et al. (2003). Pure ammonium
sulfate was used to determine the relative ionization efficiency (RIE) of sulfate
(Setyan et al., 2012). Quantification of $r$BC was calibrated using Regal Black
(REGAL 400R pigment black, Cabot Corp.) particles according to the procedures
reported in Onasch et al. (2012). Note that the solution of Regal Black was sonicated
during calibration to maintain a relative stable aerosol flow. RIEs of ammonium,
nitrate, sulfate, chloride, organics and $r$BC were determined to be 3.15, 1.05, 1.20, 1.3,
1.4 and 0.33, respectively. On the other hand, particle sizing was calibrated using
standard polystyrene latex (PSL) spheres (Duke Scientific Corp., Palo Alto, CA, USA)
across 100 - 700 nm range. Flow rate was also calibrated prior to the measurement.

Concentrations of gaseous species, e.g., carbon monoxide (CO) (Model T300,

Teledyne API, USA), ozone (O$_3$) (Model EC9810, Ecotech Pty Ltd, Australia),



nitrogen dioxide ($NO_2$) and sulfur dioxide ($SO_2$) (Model LGH-01, Anhui Landun,
China), and meteorological data including air temperature (T), relative humidity (RH),
visibility (km), wind speed (WS) and wind direction (WD) were acquired at the same
site. $PM_{2.5}$ and $PM_{10}$ mass concentrations were also recorded (BAM-1020, Met One
Instruments, Inc., USA), in parallel with the SP-AMS measurement.
**2.2 Data treatment and source analyses**
The SP-AMS data were post-processed by using the Igor-based standard
ToF-AMS Analysis Toolkit SQUIRREL v1.56D and PIKA v1.15D, available at:
http://cires1.colorado.edu/jimenez-group/ToFAMSResources/ToFSoftware/index.htm
l. Note all mass concentrations reported here were calculated from the HR fitted
results on V-mode data. A constant collection efficiency (CE) of 0.5 was used for the
mass quantification, in consistent with many other AMS studies, as indeed the mass
fraction of ammonium nitrate (mostly <40%), particle acidity (near neutral) and RH
(<10%) do not affect the CE significantly for this dataset (Middlebrook et al., 2012).
Unless specified, the concentrations of ammonium, sulfate, nitrate, chloride and
organics are from M2 setting (tungsten vaporizer only), while the *r*BC data is from
M1 setting (dual-vaporizers: tungsten + laser) in this paper. The meteorological data
(RH, T, WS, WD and visibility), concentrations of gas-phase species (CO, $NO_2$, $SO_2$
and $O_3$) and $PM_{2.5}$ were averaged into hourly data for comparisons with the SP-AMS
data. The data reported are at local time, e.g., Beijing (BJ) Time.
Positive matrix factorization (PMF) (Paatero and Tapper, 1994) was applied on
the HRMS of organic aerosol (OA) obtained under laser off W-mode (M4 setting) to
elucidate the OA sources/processes. We used the PMF Evaluation Tool version 2.08A
(downloaded     from:
http://cires1.colorado.edu/jimenez-group/wiki/index.php/PMF-AMS_Analysis_Guide)
(Ulbrich et al., 2009) to investigate the PMF results by varying the number of factors
(from 2 to 8 factors) and rotations ("*f*peak", from -1 to 1 with an increment of 0.1).
Only ions with *m/z* less than or equal to 180 were included in the analyses. Following
the instruction detailed by Zhang et al. (2011), the 4-factor solution (at *f*peak = -0.1)





was chosen as the optimal solution, as the 3-factor solution cannot separate the
hydrocarbon-like OA (HOA) and cooking OA (COA) (Fig. S2), and the 5-factor
solution clearly splits the semi-volatile oxygenated OA (SV-OOA) factor into two
OOA factors (Fig. S3). A summary of the key diagnostic plots are provided in Fig. S4.
Detailed discussion of the PMF results is presented in Section 3.5. Note we found no
significant differences between the PMF source apportionment results from the
HRMS of OA obtained with dual-vaporizers setting (M3 setting) and current results
(M4 setting, tungsten vaporizer only), as the OA HRMS acquired under these two
circumstances were overall very similar (details in Section 3.4).

**3. Results and discussion**
**3.1 Mass concentrations, chemical compositions and diurnal changes**

The temporal variations of meteorological parameters, concentrations of the gas

pollutants, concentrations and mass fractions of different $PM_1$ components, and the
$PM_{2.5}$ mass loadings (from Met one BAM-1020) over the sampling period are
illustrated in Fig. 1. During this study, the mean temperature was 18.5 $^{\circ}C$, RH on
average was 64%, and wind predominantly blew from southeast and southwest (Fig.
S5). The SP-AMS $PM_1$ concentrations ranged from 5.1 to 97.9 $\mu g\ m^{-3}$, with an
average of 28.2 $\mu g\ m^{-3}$. Note this average $PM_1$ concentration is significantly lower
than those observed during summer (38.5 $\mu g\ m^{-3}$), autumn (46.4 $\mu g\ m^{-3}$) and winter
(89.3 $\mu g\ m^{-3}$) (Zhang et al., 2015;Zhang et al., 2016b), showing that the air during
springtime in Nanjing is cleaner than in other seasons. The variations of $PM_1$
concentrations also match very well with $PM_{2.5}$ concentrations (Pearson's $r^2 = 0.72$),
and on average $PM_1$ accounts for ~ 54% of the $PM_{2.5}$ mass.

The average $PM_1$ composition is shown in Fig. 2a. The most abundant component

is found to be organics (45.0%), following by sulfate (19.3%), nitrate (13.6%),
ammonium (11.1%), $r$BC (9.7%) and chloride (1.3%). Fig. 2b further shows changes
of the $PM_1$ chemical compositions in different concentration bins. It can be seen that
although most $PM_1$ mass loadings are within 10 - 40 $\mu g\ m^{-3}$, high loading periods tend



to have higher mass contributions from organics and $r$BC, and less contributions from
secondary inorganic species, indicating that high PM events were influenced
significantly by local fresh emissions.
The molar ratio of inorganic anions (sulfate, nitrate and chloride) to cations
(ammonium) is 1.05 (Fig. 3a) (Zhang et al., 2007b). Considering that a small fraction
of sulfate, nitrate and chloride are possibly associated with metal cations, such as $Na^+$,
$K^+$ and $Ca^{2+}$, etc.,it can be concluded that the NR-$PM_1$ was overall neutral throughout
the study. On the other hand, the molar ratio of inorganic anions to ammonium is on
average 1.17 (Fig. 3b) when dual-vaporizers are on. This may be partially due to
variations of ionization/collection efficiencies of the measured species as the addition
of laser beam may change the distribution of vaporized species inside the ion chamber,
and also because of the detection of sulfate, nitrate and chloride bonded with metal
cations under the dual-vaporizers. These species don't evaporate on the tungsten
vaporizer under the laser-off mode. Indeed, more metal signals were observed with
the dual-vaporizers, as shown in Fig. S6.
Fig. 2c shows the average diurnal changes of organics, sulfate, nitrate, chloride
and $r$BC. Sulfate concentrations are slightly higher during daytime than during
nighttime, indicating a significant contribution from photochemical reactions. Sulfate
also shows the least variations among all species, reflecting its regional behavior.
Except for sulfate, all other species present a dual-peak pattern, with one peak in early
morning and another one in early evening. The peaks of $r$BC and organics are likely
due to local traffic/cooking activities (see details in Section 3.5), while the behavior of
nitrate is likely driven by the thermodynamic gas-particle partitioning: $NH_4NO_3(p) \leftrightarrow$
$NH_3(g) + HNO_3(g)$, as it shows good anti-correlations with the diurnal changes of
temperatures ($r$ = -0.72 for nitrate $vs.$ T). Furthermore, we calculated the diurnal
variations of the equilibrium constant of $NH_4NO_3$ ($K_{p,AN}$) (Young et al., 2015;Seinfeld
and Pandis, 2006) in Fig. 2c. The $K_{p,AN}$ displays a similar trend as nitrate ($r$ = 0.68),
providing strong evidence that nitrate variations were governed mainly by the
thermodynamic equilibrium. Chloride shows similar behavior as nitrate, indicating it



is driven by the equilibrium $NH_4Cl(p) \leftrightarrow NH_3(g) + HCl(g)$, as well ($r$ = -0.76 for
chloride *vs.* T). Therefore, when temperature rises, more $NH_4NO_3$ and $NH_4Cl$ can
dissociate into gaseous $NH_3$, $HNO_3$ and HCl, mass loadings of particle-phase nitrate
and chloride decrease correspondingly, and *vice versa*.

In order to further elucidate the formation processes of nitrate and sulfate, we

calculated the oxidation ratios of sulfur ($f_S$) and nitrogen ($f_N$) (Fig. 4a and 4b), defined
as $f_S = nSO_4^{2-}/(\ nSO_4^{2-} + nSO_2)$ and $f_N = nNO_3^-/(\ nNO_3^- + nNO_2)$ (Xu et al., 2014),
indicating the conversion of $SO_2$ and $NO_2$ to sulfate and nitrate, respectively. Here
$nSO_4^{2-}$, $nNO_3^-$, $nSO_2$ and $nNO_2$ are the molar quantities of particle-phase sulfate and
nitrate, gas-phase $SO_2$ and $NO_2$, respectively. Diurnal variations of $f_S$, $f_N$ and mass
ratios of $SO_4^{2-}/NO_3^-$ are presented in Fig. 4d-f, along with the diurnal cycle of RH.
The $f_S$ reaches a maximum around 3 pm; similarly, the $SO_4^{2-}/NO_3^-$ ratios are elevated
significantly during daytime, in particular during afternoon. These behaviors suggest
the remarkable role of photochemical processing of $SO_2$ to sulfate. In addition, the
diurnal profile of $f_S$ shows a negative correlation with that of RH ($r$ = -0.52),
indicating somewhat insignificant influence of aqueous-phase production of sulfate
during this campaign. Interestingly, during nighttime (7 pm – 6 am), variations of $f_N$
follows the changes of RH, probably suggesting a nighttime formation pathway of
nitrate, e.g., $N_2O_5 + H_2O = 2HNO_3$ and $HNO_3 + NH_3 = NH_4NO_3$; while the afternoon
drop of $f_N$ is likely due to evaporation of nitrate as the temperature increases.

**3.2 Chemically-resolved size distributions**

The campaign-averaged mass-based size distributions, fractional contributions

and diurnal size distributions (image plots) of the major $PM_1$ species are depicted in
Fig. 5 (temporal variations of the mass-based size distributions of these $PM_1$ species
over the whole measurement period are provided in Fig. S7). Note the size
distribution of $r$BC in these plots were scaled from the size distribution of *m/z* 24
($C_2^+$), as other major $r$BC ion clusters may be heavily influenced by other ions, such
as $C^+$ signal but from organics at *m/z* 12 ($C^+$), $HCl^+$ signal at *m/z* 36 ($C_3^+$), $SO^+$ signal





at $m/z$ 48 ($C_4^+$), $C_2H_4O_2^+$ signal at $m/z$ 60 ($C_5^+$). As can be expected, all inorganic
species (sulfate, nitrate, chloride and ammonium) display a unimodal distribution with
an accumulation mode peaking ~550 nm (vacuum aerodynamic diameter, $D_{va}$
(DeCarlo et al., 2004)), since they were mainly formed from secondary reactions. The
organics has a much broader size distribution across from ultrafine (<100 nm) to
supermicron meter range, with a small sub-peak centering ~120 nm in addition to the
major peak at ~440 nm, indicating influences from both primary and secondary
emissions. On the contrary, size distribution of $r$BC behaves very differently from
other components, which peaks at 90 - 200 nm range, reflecting clearly that it is
mainly originated from primary emissions. Overall, the small particles are
predominantly consisted of organics and $r$BC, which together account for more than
90% of the ultrafine particle mass. Mass contributions from inorganic species increase
significantly with the increase of particle size, and they dominate masses of particles
larger than 400 nm (Fig. 5b).

In line with the diurnal mass loadings of the $PM_1$ species shown in Fig. 2c, the

diurnal size distribution of sulfate is generally stable, with masses concentrating in the
400 - 700 nm range throughout the day (Fig. 5c); while the size distributions of nitrate,
chloride and organics present clear enhancements in the 300 - 700 nm range during
early morning and early evening due to increased mass concentrations of these species
during these two periods. The size distribution of $r$BC is also enhanced during the
morning and evening hours, but it extends to a much smaller size range (<100 nm).

**3.3 $PM_1$ contributions on visibility impairment**

In order to figure out the major species that are responsible for the visibility

degradation, here we employed the IMPROVE method to reconstruct the light
extinction coefficients ($b_{ext}$). $b_{ext}$ values are derived from the measured visibility:
$b_{ext}$=3.91/$V_s$ (Kong et al., 2015), where $V_s$ stands for the visibility (in meter). The
following IMPROVE formula (Yang et al., 2007) was used:
$b_{ext} = 3f(\text{RH})\{[(NH_4)_2SO_4] + [NH_4NO_3] + [NH_4Cl]\} + 4[\text{OM}] + 10[\text{BC}] + 1[\text{soil}] + 10$





Where $f$(RH) is a RH-dependent empirical coefficient which considers the effects of
water uptake by inorganic salts on the light extinction; the $f$(RH) values used here
were taken from Malm and Day (2001), which were regressed from the Great Smoky
data set. [$(NH_4)_2SO_4$], [$NH_4NO_3$], [$NH_4Cl$], [OM], and [BC] represent the mass
concentrations of ammonium sulfate, ammonium nitrate, ammonium chloride,
organics and black carbon directly from the SP-AMS measurements (in $\mu g \ m^{-3}$)
([$(NH_4)_2SO_4$] = 1.375*[$SO_4^{2-}$], [$NH_4NO_3$]=1.29*[$NO_3^-$] and [$NH_4Cl$] = 1.51*[$Cl^-$]).
Since the SP-AMS cannot accurately measure soil components (e.g., various
metals/metal oxides/metal salts), the term [soil] was set to zero during calculations.
By using this method, the reconstructed visibilities match reasonably well with
the measured values ($r^2$ = 0.50) as shown in Fig. 6a. Fig. 6b shows the time series of
the measured and reconstructed extinction coefficients throughout the whole sampling
period. It should be noted that, on average, the measured $PM_1$ species are only able to
explain ~44% of the light extinction. This is likely due to that: 1) as shown earlier, the
SP-AMS measured $PM_1$ only occupies ~54% of the $PM_{2.5}$ mass; 2) we didn't include
contributions from soil components, coarse particles and also some gas-phase species
(such as $NO_2$); 3) although the influences of water are included in part through $f$(RH)
for inorganic salts, the water uptake by organic species are not considered explicitly,
which can be significant especially for the SOA under high RH conditions (Duplissy
et al., 2011;Denjean et al., 2015). Indeed, as shown in Fig. 6a, reconstructed
visibilities appear to deviate more significantly from the measured visibilities under
high RH than ones under low RH conditions, suggesting the importance of
particle-bounded water on visibility degradation. The pie chart in Fig. 6b presents the
average relative contributions of different components to the light extinction of $PM_1$.
The largest contributor is organics which accounts for 37.7%, followed by ammonium
sulfate (25.1%), $r$BC (20.7%), ammonium nitrate (15.1%) and a minor contributor of
ammonium chloride (1.4%).

**3.4 Chemical characteristics of OA**





The unique laser vaporizer of SP-AMS allows it to detect $r$BC and species coated
on the $r$BC core including both non-refractory and refractory organics, thus
comparison between the OA mass spectra obtained with dual-vaporizers and tungsten
vaporizer settings, can infer some information regarding the chemical features of
refractory organics, that were unable to be determined by any other types of AMS. As
shown in Fig. 7a and 7b, the OA obtained with dual-vaporizers setting have slightly
higher oxygen-to-carbon (O/C) ratio (0.28 $vs.$ 0.27), nitrogen-to-carbon (N/C) ratio
(0.033 $vs.$ 0.032) and lower hydrogen-to-carbon (H/C) ratio (1.50 $vs.$ 1.52) than the
corresponding elemental ratios of OA obtained with the tungsten vaporizer only. This
result indicates that refractory organics are likely more oxygenated than the
non-refractory organics, and for this dataset it is mainly due to a higher fractional
contribution from $C_2H_3O^+$ (see the inset of Fig. 7a). This is different from the results
on laboratory-generated nascent soot, where larger $fCO_2^+$ (i.e., the fraction of total
organic signal contributed by $CO_2^+$) was observed with the dual-vaporizers setting,
indicating the variability of the chemical compositions of refractory organics. Note
the elemental ratios shown throughout the paper were all calculated based on the
method proposed by Aiken et al. (2008) (referred to as A-A method). Recently,
Canagaratna et al. (2015) improved this methodology by using specific ion fragments
as markers to calculate the O/C and H/C ratios (referred to as I-A method). The I-A
method increased the O/C ratio, H/C ratio, and the OM/OC ratio higher than the
values calculated from the the A-A method, on average, by 28%, 10% and 8%,
respectively (Fig. S8). In this work, we used the results from the A-A method for
consistency and comparisons with previous AMS measurements.
Overall, the O/C ratio (0.27) of OA in Nanjing during springtime is a bit lower
than those observed at other urban locations in China – for instances, 0.30 in
Shenzhen (He et al., 2011), 0.31 in Shanghai (Huang et al., 2012b), 0.33 in Lanzhou
(Xu et al., 2014) and 0.34 in Beijing (Zhang et al., 2014), and much lower than those
at rural sites – for instances, 0.47 in Kaiping (Huang et al., 2011) and 0.59 in
Changdao (Hu et al., 2013). As O/C ratio is a good indicator of the aging degree of



OA, the relatively low O/C level indicates a significant contribution from fresh
emissions in Nanjing aerosols during springtime. Accordingly, the non-refractory OA
(pie chart in Fig. 7b) is dominated in hydrocarbon $C_xH_y^+$ ions (51.2%) rather than the
oxygen-containing ion fragments (37.4% of $C_xH_yO_1^+$ and $C_xH_yO_2^+$).

The scatter plot of $f44$ (mass fraction of $m/z$ 44 to the total OA) *vs.* $f43$ (mass

fraction of $m/z$ 43 to the total OA) (a.k.a., triangle plot) (Ng et al., 2010) was often
used to investigate the oxidation degrees of OA. As presented in Fig. 8, most OA
reside in the bottom end of the triangular region, again pointing out the
less-oxygenated behavior of the OA. Since the HRMS can separate different ions at
the nominal $m/z$, we also examined the $fCO_2^+$ *vs.* $fC_2H_3O^+$ space and illustrated it in
Fig. S9 - many OA locate outside the triangular region, yet still close to the bottom.
Moreover, $m/z$ 60 (mainly $C_2H_4O_2^+$) is a significant fragment ion of levoglucosan,
which is well known as the biomass burning aerosol tracer (Alfarra et al., 2007).
However, as $f60$ (mass fraction of $m/z$ 60 to the total OA) is very low in OA (average
$\pm 1\sigma = 0.4 \pm 0.06$ %), indicating no biomass burning influences on the OA properties
during springtime in Nanjing.

**3.5 Sources and evolution processes of OA**

In order to further elucidate the sources and evolution processes of OA, we

performed PMF analyses and identified four OA components, including two primary
OA (POA) factors – a traffic-related hydrocarbon-like OA (HOA) and a
cooking-related OA (COA), and two secondary OA factors – a semi-volatile
oxygenated OA (SV-OOA) and a low volatility OOA (LV-OOA). Details about their
characteristics are discussed below.
**3.5.1 Mass spectral features of the OA factors**

The mass spectral profiles, time-dependent mass concentrations of the four OA

factors and corresponding tracer ions are presented in Fig. 9. The HOA mass spectrum
is overall dominated by the $C_xH_y^+$ ions (73.2%), such as $C_3H_7^+$, $C_4H_7^+$, $C_4H_9^+$, $C_5H_9^+$
etc., which are most likely produced from alkanes and cycloalkanes emitted from fuel



and lubricating oil burning (Canagaratna et al., 2004). This feature is in good
agreement with the mass spectral features of POA directly from vehicle
emissions(Collier et al., 2015), and the HOA factors determined in many other
locations (e.g., Ge et al., 2012b;Huang et al., 2010;Sun et al., 2011). HOA has the
lowest O/C ratio (0.10) and highest H/C ratio (1.75) among all factors, representing its
behavior as primary fresh emissions. The COA mass spectrum is also rich in $C_xH_y^+$
ions (64.7%), but having more oxygenated ions ($C_xH_yO_z^+$) than the HOA (26.5% *vs.*
15.4%), especially $C_3H_3O^+$ and $C_3H_5O^+$ ions. The significant contributions of $C_3H_3O^+$
and $C_3H_5O^+$ to *m/z* 55 and *m/z* 57 are a common feature of COA, that has been
reported in various urban locations around the world, for examples, Beijing (Sun et al.,
2015a), London (Allan et al., 2010), Fresno (Ge et al., 2012b), New York City (Sun et
al., 2011) and Barcelona (Mohr et al., 2012;Mohr et al., 2015). These
oxygen-containing ions are in part generated from the fragmentation of fatty acids in
the cooking aerosols (Ge et al., 2012b). As a result, COA has a higher O/C ratio of
0.16 and a lower H/C ratio of 1.67 than those of HOA. The O/C and H/C levels of
COA in this work are also close to those identified in other locations aforementioned.
The consistency of the chemical characteristics of COA from such different locations
suggests that ambient COA is more relevant to the cooking oil rather than the different
types of food, which was postulated earlier by Allan et al. (2010).
Unlike the two POA factors, SV-OOA and LV-OOA are both abundant in
oxygen-containing fragments ($C_xH_yO_z^+$ ions), which are 46.4% and 54.8%,
respectively. The higher O/C ratio (0.55 *vs.* 0.32) and more $C_xH_yO_2^+$ ions (18.8% *vs.*
11.8%) in the LV-OOA mass spectrum than those of the SV-OOA, reflecting the fact
that LV-OOA went through more aging/oxidation reactions than the SV-OOA. The
O/C ratio of SV-OOA is 0.32, which is within the O/C range of SV-OOA observed
worldwide (Jimenez et al., 2009). The LV-OOA O/C ratio of 0.55 is in the lower end
compared to the O/C levels of LV-OOA observed in other China sites, for examples,
0.64 in Kaiping (Huang et al., 2011), 0.65 in Shanghai (Huang et al., 2012b), 0.68 in
Lanzhou (Xu et al., 2014), 0.78 in Changdao (Hu et al., 2013) and 0.80 in Hong Kong



(Lee et al., 2013).

Consistently, in the $f44$ *vs.* $f43$ space (Fig. 8), SV-OOA situates near the bottom

side while LV-OOA approaches to the upper part of the triangular region, because of a
much larger fractional contribution of $CO_2^+$ in the LV-OOA mass spectrum. HOA and
COA, as POA factors, both reside in the bottom end of the plot, away from SV-OOA
and LV-OOA; while they locate outside the triangle in the $fCO_2^+$ *vs.* $fC_2H_3O^+$ space
(Fig. S9), indicating that the HRMS acquired by the SP-AMS is better in
differentiating POA factors from other SOA factors than the unit mass resolution
(UMR) data.

In order to justify the OA factors identified in this study, we compared the

spectral similarities of the OA factor spectral profiles (in both HR and UMR) with
those separated during wintertime in Beijing (Sun et al., 2015a), summertime in
Lanzhou (Xu et al., 2014), and wintertime in Fresno (Ge et al., 2012b;Ge et al.,
2012a). The results are listed in Table 1. Indeed, the HOA, COA and LV-OOA mass
spectra are highly similar to the corresponding factors identified in Bejing, Lanzhou
and Fresno ($r^2 > 0.87$); SV-OOA also correlates fairly well with Bejing and Lanzhou
SV-OOA too, but with relative low $r^2$ (0.68 – 0.75), mainly because of one or two ion
fragments, namely, higher $CO^+$ and $CO_2^+$ signals in Beijing SV-OOA and higher
$C_2H_3O^+$ signal in Lanzhou SV-OOA than those in Nanjing SV-OOA. The SV-OOA on
the other hand, correlates very well with the Fresno OOA ($r^2 = 0.90$ and 0.91).

Moreover, as presented in Fig. 9a, the HOA mass spectrum contains relatively

higher fraction of ions with large *m/z* values (*m/z* > 100) than that of COA (14.0% *vs.*
8.2%), and most of these ions are $C_xH_y^+$ ions, probably from fuel burning emitted
long-chain alkanes, etc. The SV-OOA also includes more large *m/z* ion fragments (*m/z*>
100) than those in the LV-OOA mass spectrum (10.5% *vs.* 5.3%), likely suggesting
that further oxidation of SOA species may lead to the fragmentation of high molecular
weight species and formation of small molecules – a mechanism verified by both
lab-scale experiments (e.g., Yu et al., 2014) and field measurements (e.g., Lee et al.,

2012).



### 3.5.2 Temporal variations, diurnal patterns and relative contributions of the OA factors


The temporal variations of different OA factors and their corresponding tracer
ions are displayed in Fig. 9b. $C_4H_9^+$ ion, a.k.a., the HOA mass spectral tracer (Zhang
et al., 2005) indeed varies very closely to the HOA ($r^2 = 0.94$). Time series of the COA
tracer ion $C_6H_{10}O^+$ (and also $C_5H_8O^+$, $C_7H_{12}O^+$) (Sun et al., 2011;Ge et al., 2012b)
match very well with that of COA too ($r^2 = 0.90$). SV-OOA correlates better with
$C_2H_3O^+$ ($r^2 = 0.90$) than with $CO_2^+$ ($r^2 = 0.66$). Although LV-OOA doesn't correlate
very well with $CO_2^+$ ($r^2 = 0.12$) mainly due to the mismatch during April 23 - 26, the
correlation is still much better than it with $C_2H_3O^+$ ($r^2 < 0.001$). In Table 2, we
tabulate the correlation coefficients ($r$) of the four OA factors with the gas-phase
species, BC and inorganic species. Note we used Pearson's $r$ not $r^2$ here since some
correlation coefficients are negative. From the table, it is clear that the traffic-related
gaseous species, CO and $NO_2$, correlate best with HOA among all OA factors;
SV-OOA correlates better with nitrate ($r = 0.49$) than it with sulfate ($r = 0.11$);
LV-OOA correlate better with sulfate ($r = 0.23$) that it with nitrate ($r = 0.11$). All these
results are consistent with the traffic origin of HOA, the semi-volatile and
low-volatility behaviors of SV-OOA and LV-OOA.
Accordingly, diurnal cycles of the OA factors are presented in Fig. 10a.
Correlation coefficients ($r$) of the diurnal variations between OA factors with
gas-phases and inorganic species are provided in Table 2, as well. HOA
concentrations show an early morning peak, and it overall remains at high levels
during nighttime. Besides the impacts of boundary layer height, this is also due to
enhanced emissions from construction vehicles around the site, which were in fact
much more active during nighttime than during daytime because of the restrictions of
Nanjing government. Most of those vehicles used low-quality diesel fuel, and could
emit a large amount of $r$BC particles. The $r$BC diurnal pattern is indeed almost
identical to that of HOA ($r = 0.99$), indicating that the HOA during this campaign was
apparently associated with the construction vehicle emissions. COA concentrations



increase during noon (12 pm – 1 pm) and early evening, in response to the lunchtime
and dinnertime cooking activities. SV-OOA concentrations decreases from 9 am, and
reach a minimum during afternoon (3 pm – 4 pm), oppositely to the variation of
temperatures ($r = -0.85$) but similar to that of nitrate ($r = 0.53$), corroborating its
semi-volatile feature. Different from other factors, LV-OOA concentrations increase
during daytime and shows positive correlation with temperature ($r = 0.76$); it also has
negative correlation with the diurnal cycle of RH ($r = -0.75$). Both behaviors are
similar to those of sulfate ($r = 0.72$ for the diurnal cycle of LV-OOA *vs.* sulfate),
indicating the leading role of photochemical oxidation for LV-OOA formation as well.
As shown in Fig. 10b, due to mainly the increase of LV-OOA mass loading, OA is
overwhelmingly dominated by the SOA (SV-OOA + LV-OOA) during afternoon (80.2%
at 3 pm); POA (HOA + COA) only dominates the OA mass during morning (53.2% at
7 am) and early evening (56.9% at 8 pm) in response to the enhanced traffic and
cooking emissions. On average, the OA is composed of 27.6% of HOA, 16.9% of
COA, 27.4% of SV-OOA and 28.1% of LV-OOA (Fig. 10c), with SOA outweighing
POA (55.5% *vs.* 44.5%). However, as shown in Fig. 10d, with the increase of OA
mass loadings, the fractional contribution of POA increases, highlighting the
important and direct influences of anthropogenic emissions on the heavy pollution
haze events.

**3.5.3 Local/regional influences and evolution processes of the OA factors**
Combining WS, WD and mass loadings, the bivariate polar plots of the four OA
factors, $r$BC, and total PM$_1$ are shown in Fig. 11. These plots provide an effective
graphical method for showing the potential influences of air masses from different
directions with different wind speeds to the receptor site (Carslaw and Beevers, 2013).
Clearly, high mass loadings of HOA and $r$BC mostly link with low WS ($< 1$ m s$^{-1}$),
indicating they are mainly from local vehicle emissions. High COA concentrations
occur mainly under low WS as well, but with some high concentrations accompanied
with air masses from southeast under higher WS. SV-OOA appears to be formed





locally, except for a concentration hotspot in the southeast – likely due to emissions
from the tobacco factory that resides in that direction. High concentrations of
LV-OOA are distributed in all directions under higher WS, representing its regional
behaviors. Overall, high $PM_1$ mass loadings occur mainly under low WS, indicating
that the $PM_1$ is heavily affected by local emissions rather than pollutants in a regional
scale.

The aging of OA can be described in general by the increase of O/C and decrease

of H/C. In this regard, we plotted the Van Krevelen diagram (Heald et al., 2010) (Fig.
12a) to show the relationships between H/C and O/C ratios for all OA as well as the
four OA factors. Overall, in this study, the H/C and O/C ratios of OA data are
correlated linearly with a slope of -1.04 ($r^2 = 0.93$), indicating the propagation of OA
is similar to an aging process that is likely driven by the addition of carboxylic acid
(slope of -1). Interestingly, the two OOA factors lie very well on the fitted straight line.
This trend may suggest that the evolution of secondary OA during this campaign
follows a transformation pathway of SV-OOA to LV-OOA through the addition of
carboxylic acid. The diurnal cycle of LV-OOA varies oppositely to that of SV-OOA ($r$
= -0.86), probably supporting this hypothesis. In addition, O/C ratios of OA show no
obvious correlation with the RH as shown in Fig. 12b, verifying that aqueous-phase
processing is insignificant compared to the photochemical processing for the
oxidation of OA.

**4. Conclusions**

We present for the first time the real-time measurement results using the SP-AMS

on submicron aerosols in urban Nanjing during springtime (April 13 - 29, 2015). The
SP-AMS determined $PM_1$ mass loadings, agreed well with the $PM_{2.5}$ concentrations
measured by the Met One $PM_{2.5}$ analyzer. The average $PM_1$ concentration was 28.2 μg
$m^{-3}$, lower than previously ACSM-determined $PM_1$ concentrations during summer and
winter in Nanjing. Organics on average comprised the largest fraction (45%) of $PM_1$,
and its fractional contributions increased in case of high $PM_1$ mass loadings. The





diurnal cycles of mass concentrations of organics, $r$BC, nitrate and chloride all
presented a similar behavior, which was high in early morning and evening, but low
in the afternoon. Concentrations of sulfate, on the contrary, increased during afternoon.
Further investigations of $f_S$, $f_N$, $SO_4^{2-}/NO_3-$ and RH revealed that photochemical
processing contributed significantly to sulfate formation, while nitrate (and chloride)
formation was mainly governed by the thermodynamic equilibrium. The
chemically-resolved mass-based size distribution data showed that $r$BC occupied a
large fraction of ultrafine particles, while secondary inorganic species could dominate
the mass of particles larger than 400 nm ($D_{va}$). In addition, by using the IMPROVE
method, we found that the observed $PM_1$ components were able to reproduce ~44% of
the light extinction during this study.
PMF analyses resolved four OA factors, e.g., HOA, COA, SV-OOA and LV-OOA.
Mass spectral profiles of these factors agree very well with the corresponding factors
identified at other locations. The springtime OA showed no influences from biomass
burning emissions. On average, the OA is dominated by SOA (55.5%), but POA
appeared to be more important when the OA mass loadings are high, and can be
dominant in early morning and evening. Diurnal cycle of SV-OOA varied similarly to
that of nitrate, reflecting its semi-volatile behavior. Diurnal variations of LV-OOA
showed great resemblance to that of sulfate, indicating its formation was mainly from
photochemical oxidation, as well. The bivariate polar plots indicate that SV-OOA was
formed locally, and the Van Krevelen diagram further suggests a transformation
pathway of SV-OOA to LV-OOA probably via the addition of carboxylic acid.
Generally, our highly time-resolved SP-AMS measurement results may offer useful
insights into the aerosol chemistry, and have important implications for the PM
control and reduction in this densely populated region.

**Acknowledgements**
This work was supported by the Natural Science Foundation of China (Grant Nos.
21407079 and 91544220), the Jiangsu Natural Science Foundation (BK20150042),





the Jiangsu Provincial Specially-Appointed Professors Foundation, the Jiangsu
Innovation and Entrepreneurship Program, the Startup Foundation for Introducing
Talent of NUIST (2014r064), and the LAPC Open Fund (LAPC-KF-2014-06). M.
Chen also acknowledges the support from the Natural Science Foundation of China
(Grant Nos. 21577065 and 91543115), the Commonweal Program of Environment
Protection Department of China (201409027-05), and the International ST
Cooperation Program of China (2014DFA90780). The authors thank Nanjing
Environmental Monitoring Center for the supporting data, and the help from Shun Ge,
Ling Li, Yanan He, Hui Chen and Yangzhou Wu during the campaign and preparation
of the manuscript.

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

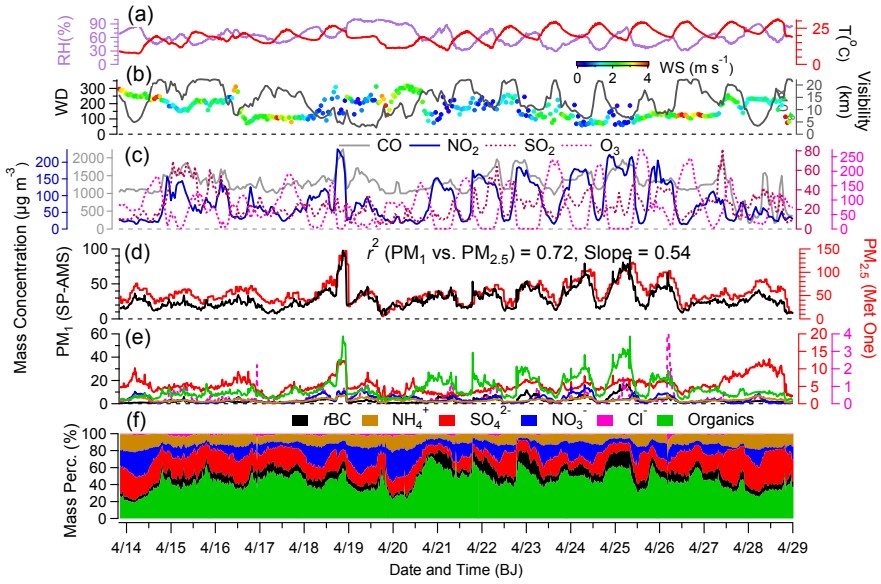


Figure 1. Time series of (a) relative humidity (RH) and temperature (T), (b) wind direction (WD) colored by wind speed (WS, m s$^{-1}$) and visibility (km), (c) mass concentrations of CO, NO$_2$, SO$_2$ and O$_3$ (hourly data), (d) mass concentrations of PM$_1$ measured by the SP-AMS, and PM$_{2.5}$ measured by the co-located Met One PM$_{2.5}$ analyzer, (e) mass concentrations of $r$BC, ammonium, sulfate, nitrate, chloride and organics, and (f) mass contributions (%) of the six PM$_1$ components (BJ, Beijing).

939

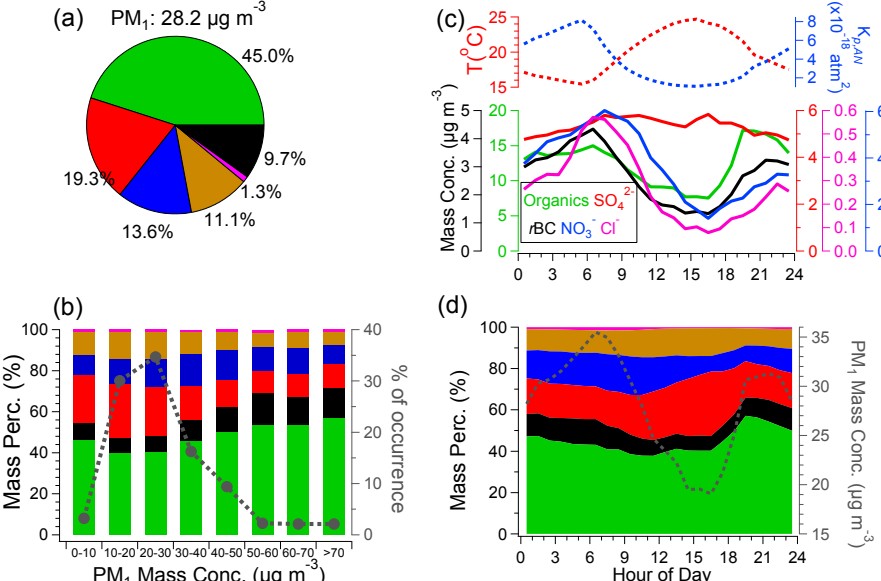

Figure 2. (a) Campaign-averaged mass contributions of organics, sulfate, nitrate, ammonium, chloride and $r$BC to the total $PM_1$, (b) mass percentages of the six $PM_1$ species (left $y$-axis) and, fractions of the number of data points to the total number of data points for $PM_1$ at different concentration bins (right $y$-axis), (c) diurnal patterns of mass concentrations of the major $PM_1$ species (bottom panel), temperature (top panel, left $y$-axis), and the equilibrium constant ($K_{p,AN}$) of $NH_4NO_3$ (top panel, right $y$-axis) ($K_{p,AN} = K_{p,AN}(298) exp \left\{ a \left( \frac{298}{T} - 1 \right) + b \left[ 1 + ln \left( \frac{298}{T} \right) - \frac{298}{T} \right] \right\}$, for reaction $NH_3(g) + HNO_3(g) \leftrightarrow NH_4NO_3(p)$. $K_{p,AN}(298)$ is the equilibrium constant at 298 K ($3.36 \times 10^{-16}$ atm$^2$), $a = 75.11$, and $b = -13.5$ (Seinfeld and Pandis, 2006)), (d) diurnal variations of mass fractional contributions of the six $PM_1$ species (left $y$-axis), and the $PM_1$ mass concentrations (right $y$-axis).



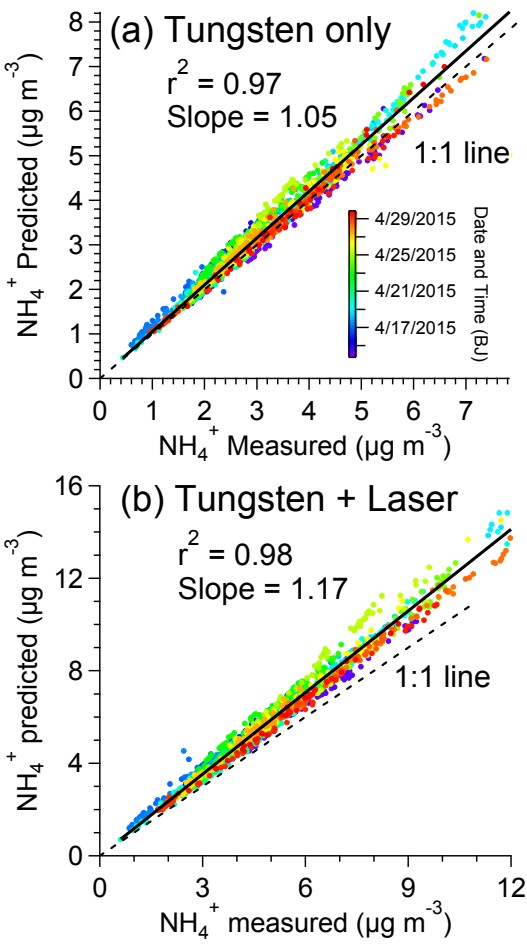

955

Figure 3. Scatter plots of the predicted $NH_4^+$ *vs.* measured $NH_4^+$ concentrations
(colored by time), in the case of (a) tungsten vaporizer only, and (b) dual-vaporizers
(tungsten + laser). The predicted values were calculated according to the formula:
$NH_4^+$ predicted = $18 \times (2 \times SO_4^{2-}/96 + NO_3^-/62 + Cl^-/35.5)$ (Zhang et al., 2007b).

960

961



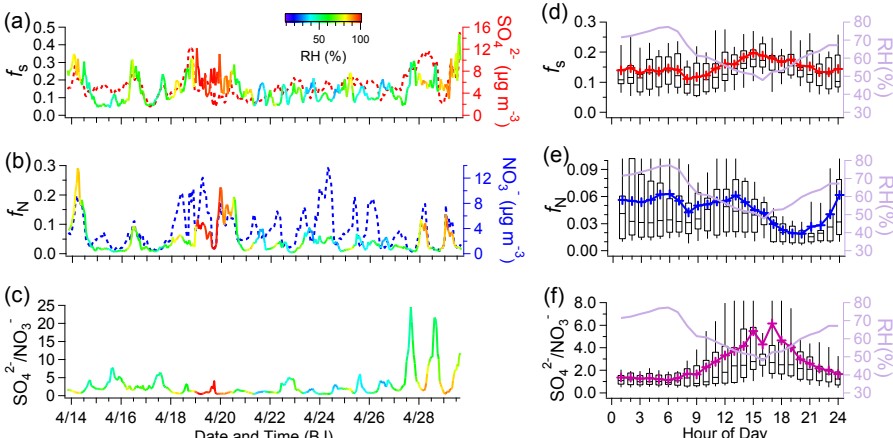

Figure 4. Time series of (a) sulfur oxidation ratio, $f_S = nSO_4^{2-}/(nSO_4^{2-} + nSO_2)$, and sulfate, (b) nitrogen oxidation ratio, $f_N = nNO_3^-/(nNO_3^- + nNO_2)$, and nitrate, and (c) mass ratios of sulfate to nitrate ($f_S$, $f_N$ and $SO_4^{2-}/NO_3^-$ are colored by the relative humidity (RH) values), diurnal variations of (d) $f_S$, (e) $f_N$, and (f) $SO_4^{2-}/NO_3^-$ and RH (the lines and cross symbols indicate the mean values, the lines in the boxes indicate the median values, the upper and lower boundaries of the boxes indicate the 75th and 25th percentiles, and the whiskers above and below the boxes indicate the 90th and 10th percentiles).



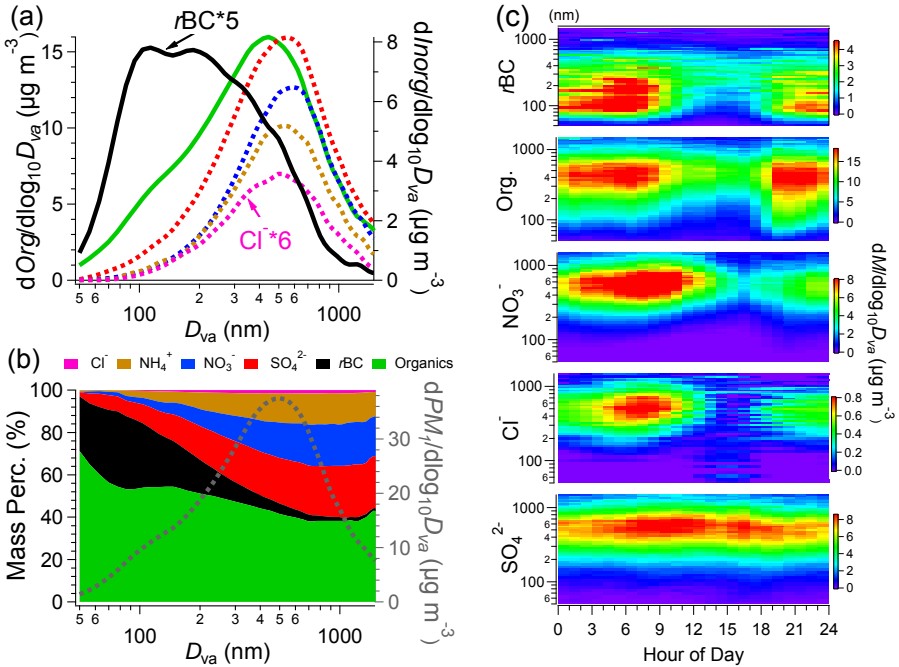

Figure 5. (a) Mass-based average size distributions of organics, $r$BC (left $y$-axis),
sulfate, nitrate, chloride and ammonium (right $y$-axis) ($D_{va}$, vacuum aerodynamic
diameter), (b) fractional contributions of the six $PM_1$ species as a function of particle
size (left $y$-axis), and size distribution of total $PM_1$ (right $y$-axis), (c) diurnal profiles
of the size distributions of $r$BC, organics, nitrate, chloride and sulfate.





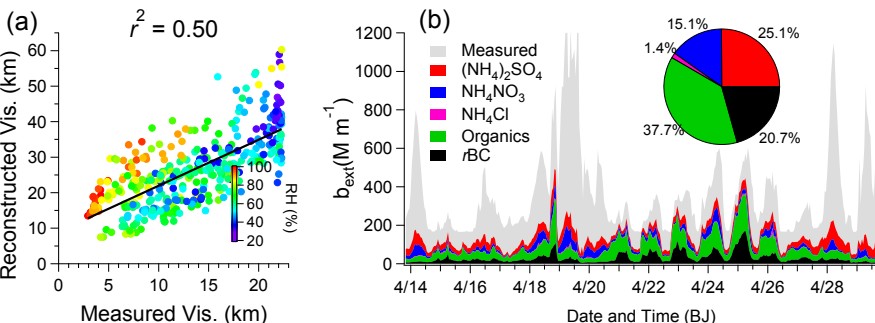


Figure 6. (a) Scatter plot of reconstructed *vs.* measured visibility (colored by RH), (b)
light extinction coefficients derived from measured visibility (grey), and reconstructed
from SP-AMS measured ammonium sulfate, ammonium nitrate, ammonium chloride,
organics and *r*BC using the IMPROVE method. The inset pie shows the relative
contributions of the five species to the light extinction of $PM_1$.






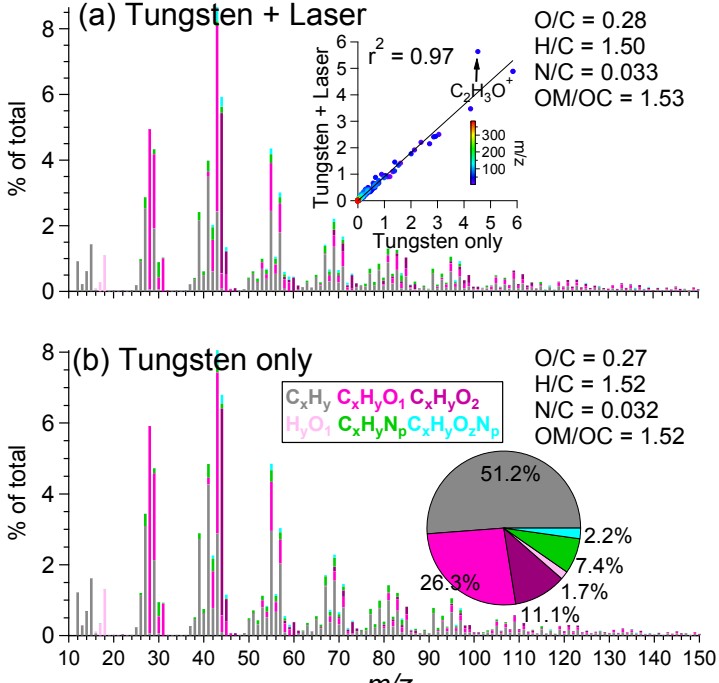


Figure 7. Campaign-averaged high resolution mass spectra of OA colored by six ion
categories, in the case of (a) dual-vaporizers (tungsten + laser) (the inset scatter plot
compares the spectral similarity between (a) and (b)), and (b) tungsten vaporizer only
(the inset pie shows the relative contributions of six ion categories to the total OA).




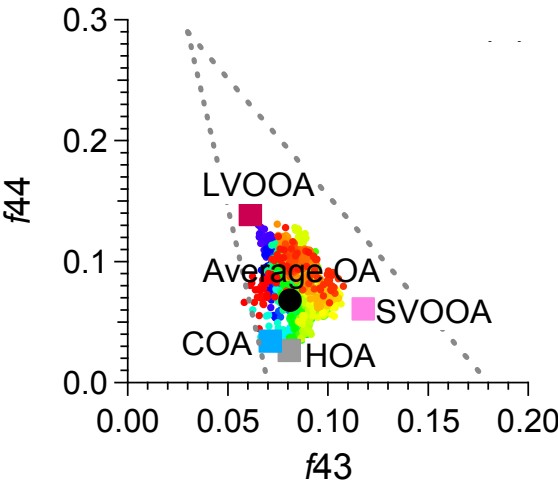


Figure 8. Triangle plot of *f*44 *vs.* *f*43 for all OA (colored by time), and the four OA
factors identified by the PMF analyses.







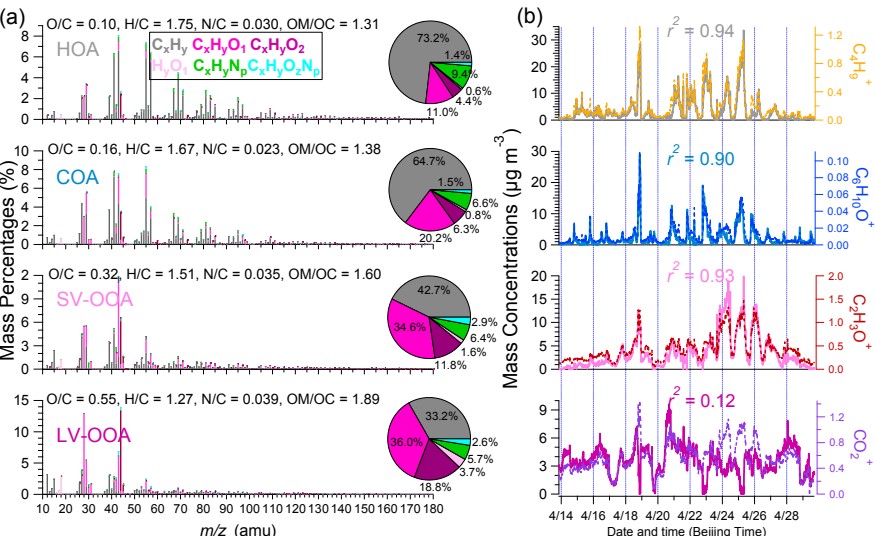


Figure 9. (a) High resolution mass spectra of hydrocarbon-like OA (HOA), cooking-related OA (COA), semi-volatile oxygenated OA (SV-OOA), and low volatility oxygenated OA (LV-OOA) colored by six ion categories (the four inset pies show the relative contributions of the six ion categories to the four OA factors, respectively), (b) time series of the four OA factors and corresponding tracer ions.



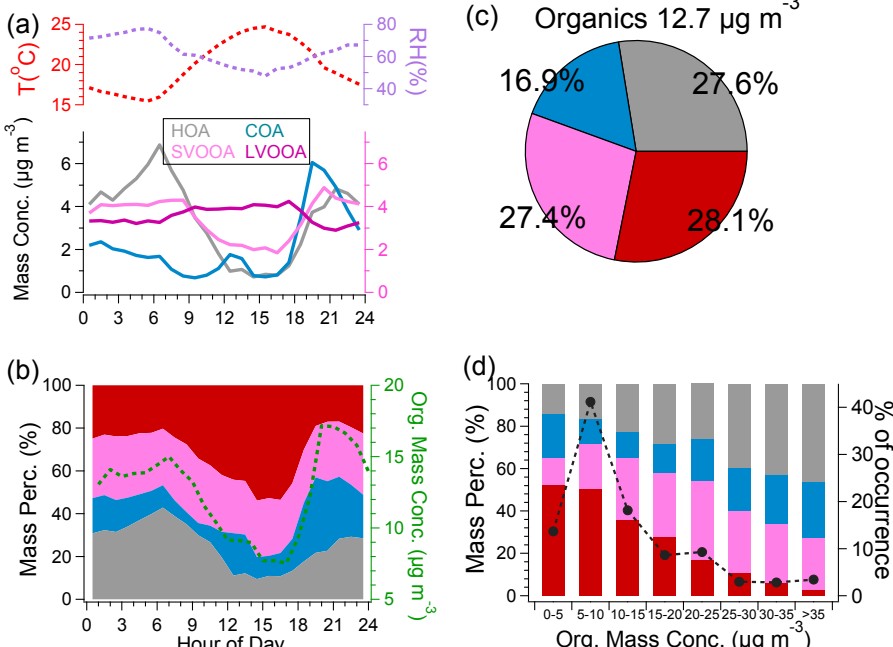

Figure 10. (a) Diurnal cycles of mass concentrations of the four OA factors (bottom
panel), temperature (top panel, left *y*-axis) and RH (top panel, right *y*-axis), (b) diurnal
variations of mass contributions of the four OA factors (left *y*-axis), and the total OA
mass concentrations (right *y*-axis), (c) campaign-averaged mass contributions of the
four OA factors to the total OA mass, and (d) mass contributions of the four OA
factors (left *y*-axis), and the fractions of the number of data points to the total number
of data points for the OA at different concentration ranges (right *y*-axis).





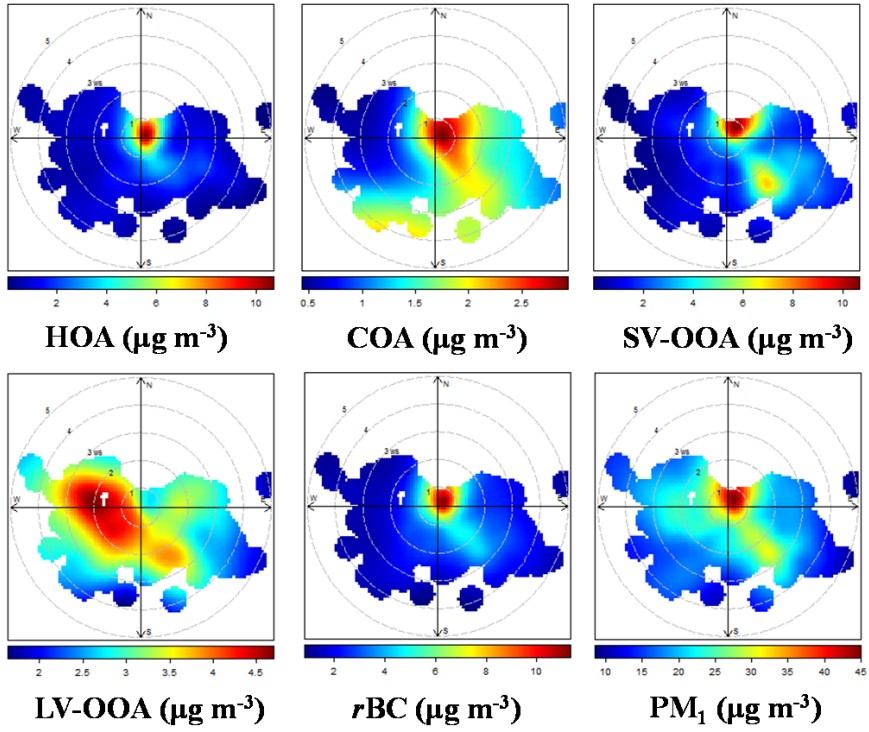

HOA (µg m⁻³)          COA (µg m⁻³)          SV-OOA (µg m⁻³)

LV-OOA (µg m⁻³)          *r*BC (µg m⁻³)          PM₁ (µg m⁻³)

Figure 11. Bivariate polar plots of HOA, COA, SV-OOA, LV-OOA, $r$BC and PM$_1$
(the color scale shows the concentration of each species, and the radical scale shows
the wind speed that increases outward from the center).






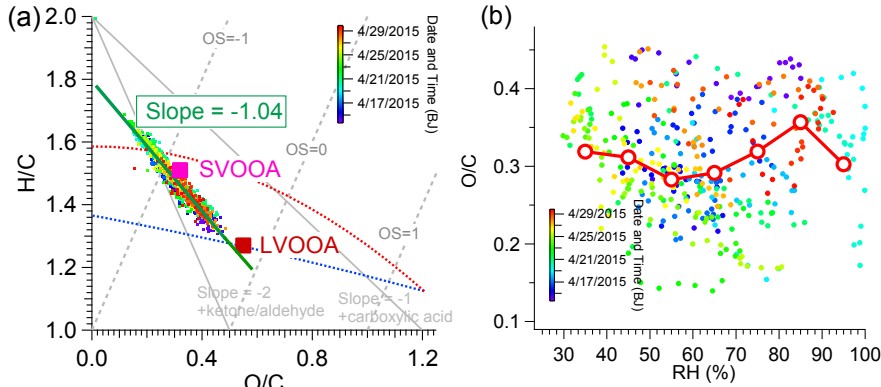


Figure 12. (a) Van Krevelen diagram of H/C *vs.* O/C ratios for all OA data colored by
time, the blue and red dashed lines correspond to the right and left grey dashed lines
in the *f*44 vs. f43 triangle plot of Fig. 8, the grey lines represents the addition of a
particular functional group to an aliphatic carbon (Heald et al., 2010), (b) scatter plot
of O/C *vs.* RH (colored by time), the circles represents the average O/C values of the
RH bins (10% increment).





Table 1. Correlation coefficients (Pearson's $r^2$) between the mass spectral profiles of
the OA factors identified in this work with the corresponding factors identified in
Beijing (2013 Winter) (Sun et al., 2015a), Lanzhou (2014 Summer) ((Xu et al., 2014)),
and Fresno (2010 Winter) (Ge et al., 2012b).

| Nanjing (2015 Spring) | High resolution MS ($r^2$) | | |
|---|---|---|---|
| | Beijing (2013 Winter) | Lanzhou (2012 Summer) | Fresno (2010 Winter)* |
| HOA | 0.92 | 0.90 | 0.98 |
| COA | 0.93 | 0.94 | 0.93 |
| SV-OOA | 0.68 | 0.75 | 0.90 |
| LV-OOA | 0.91 | 0.98 | 0.87 |
| | Unit mass resolution MS ($r^2$) | | |
| HOA | 0.92 | 0.91 | 0.99 |
| COA | 0.96 | 0.96 | 0.95 |
| SV-OOA | 0.70 | 0.74 | 0.91 |
| LV-OOA | 0.90 | 0.98 | 0.89 |

*Note the Fresno (2010 Winter) study only identified one OOA factor, we thus
compared both SV-OOA and LV-OOA in this study with it.





Table 2. Correlation coefficients (Pearson's $r$) between the time series of the four OA
factors with the gas-phase species (hourly data) and other $PM_1$ components (15-min
data), and the correlation coefficients between the diurnal data.

| Pearson's $r$ | Temp.(T) | CO | $NO_2$ | $SO_2$ | $O_3$ | $SO_4^{2-}$ | $NO_3^-$ | $Cl^-$ | $r$BC |
|---|---|---|---|---|---|---|---|---|---|
| | | | Hourly data | | | | 15-min data | | |
| HOA | -0.14 | 0.71 | 0.77 | 0.13 | -0.54 | 0.15 | 0.26 | 0.45 | 0.92 |
| COA | 0.11 | 0.50 | 0.58 | -0.06 | -0.22 | 0.19 | 0.07 | 0.08 | 0.61 |
| SVOOA | 0.19 | 0.41 | 0.70 | 0.14 | -0.21 | 0.11 | 0.49 | 0.25 | 0.70 |
| LVOOA | 0.069 | -0.2 | -0.18 | 0.06 | 0.14 | 0.23 | 0.11 | 0.01 | -0.22 |
| | | | Diurnal data | | | | | | |
| HOA | -0.94 | 0.86 | 0.86 | 0.66 | -0.96 | -0.35 | 0.72 | 0.82 | 0.99 |
| COA | -0.15 | 0.28 | 0.59 | -0.24 | -0.24 | -0.57 | -0.33 | -0.25 | 0.19 |
| SVOOA | -0.85 | 0.86 | 0.94 | 0.58 | -0.90 | -0.51 | 0.53 | 0.61 | 0.89 |
| LVOOA | 0.76 | -0.58 | -0.83 | -0.27 | 0.77 | 0.72 | -0.26 | -0.33 | -0.75 |

