# Peer review of "Highly time-resolved urban aerosol characteristics during springtime in Yangtze River Delta, China: Insights from soot particle aerosol mass spectrometry"

_Atmospheric Chemistry and Physics, 2016_

## Referee Comment (RC1) · Dr. Chen (Referee) · 8 Apr 2016

This manuscript reports the measurement results of submicron aerosols by the SP-AMS in Nanjing. Recently the Aerodyne AMS has been widely used around the world, and this work presents for the first time the results using the SP-AMS in the YRD region. This is overall a very well written paper with quite thorough analyses of the data, the figures are informative and the results provide new insights regarding the aerosol chemistry in this region. I recommend its publication in ACP after addressing of the following comments:

(1) As the paper is submitted to the PEEX special issue, it will be good for the authors to describe the link of the results presented in this work with the overall scientific goal of this special issue. (2) "A constant collection efficiency (CE) of 0.5 was used for the mass quantification, in consistent with many other AMS studies, as indeed the mass fraction of ammonium nitrate (mostly <40%), particle acidity (near neutral) and RH(<10%) do not affect the CE significantly for this dataset (Middlebrook et al., 2012)."It is a bit difficult for non-AMS users to understand this point. Consider rephrasing to make the statement clear. (3) In Fig.2, the authors used the KpAN values to elucidate the formation mechanism of nitrate. Do the authors have gaseous NH3 and HNO3 data to make the argument more robust? (4) In Fig.2b and Fig.10d, the authors seperated the data into bins with 5ug/m3 increment, i think it will be better to make similar plots that describes the variations of aerosol compostions continously against the mass loadings. (5) Reconstruction of the light extinction by using the IMPROVE method is interesing and valuable. However, the parameters of the IMPROVE formula used were taken from the dataset in US, are there more appropraite values that can be used? (6) It is interesting to use polar plots to demonstrate the characteristics of organic aerosols. How about similar plots for the inorganic species? (7) As the SP-AMS can measure both non-refractory and refractory species, it may provide unique information regarding the composition of refractory organics which cannot be measured by other AMS, this reviewers feel relevant discussion is lacking in the current manuscript. (8) In Fig.12, the authors show the O/C vs. RH. Instead, i suggest to show the SV-OOA and LV-OOA mass concentrations with the increase of RH, so as to better demonstrate the influences of RH on the SOA formation.

---

## Referee Comment (RC2) · Anonymous Referee #2 · 25 Apr 2016

The paper of Wang et al. describes the chemical characteristics of urban aerosol in Yangtze River Delta, China, measured by using a soot particle aerosol mass spectrometer. The results showed that most of the submicron particles consisted of organics, sulfate, nitrate, ammonium and black carbon. Authors used the results on chemical species to construct the light extinction, chemically resolved mass size distributions and the source apportionment of organics. Four factors were found for organic aerosol in Yangtze River Delta, hydrocarbon-like, cooking-related, semivolatile oxygenated and low-volatility oxygenated OA. Secondary OA dominated total organics, but when large OA concentrations were observed, the contribution of primary organics increased indicating the importance of anthropogenic sources.

This paper is well-written and the data have been analyzed and discussed very thoroughly. However, as there are so many AMS papers published on urban aerosols in past 15 years, it is difficult to see what is the significance and novelty of this paper. Are there any results presented in this paper that have not been published before? I suggest authors to think carefully what is the contribution of this paper to aerosol science and emphasize that clearly in the manuscript. Maybe it could be the unique features of the SP-AMS that allows to investigate the refractory material in addition to non-refractory species. In that case I suggest to focus this paper on that topic more clearly. This paper should be published after minor revision.

Specific comments:

1. Page 2, line 34: remove ($\sim$54% of the PM2.5 mass) unimportant detail

2. Page 5, line 124: Why springtime? Is there something specific in aerosol chemistry in springtime Yangtze River Delta? Add a reason for springtime measurements.

3. Page 6, line 148: "coated species" Specify. What are the core and what are the coating species?

4. Page 7: lines 183-186: collection efficiency, how is CE defined for the SP-AMS? Discuss with relevant citation.

5. Page 8, 206-210: no difference in OA factors between dual vaporizer and tungsten vaporizer; how about rBC? With dual vaporizer set-up you are able to separate rBC for different PMF factors. It would have been interesting to see how rBC divides between HOA and COA, or does it?

6. Page 17 and Fig 9; I suggest rather showing the correlation (time-series) of PMF factors and inorganic species than PMF factors and organic tracers in Figure 9.

7. Page 35, Figure 5a: Is it possible that the right side of the mass size distributions

is defined by the transmission of aerodynamic lens? Could you estimate how much of the accumulation mode mass is missing for organics, sulfate, nitrate, ammonium and chloride because of that?

Technical corrections:

1. Page 3, lines 67-68: parenthesis are used unclearly

2. Page 15, line 428: . . . the SV-OOA. . . remove "the" as you haven't used it with PMF factors earlier

---

## Referee Comment (RC3) · Anonymous Referee #3 · 29 Apr 2016

General comments: The paper deals with highly time-resolved urban aerosol characterisation during two weeks period in spring in Nanjing (China) using soot particle high resolution aerosol mass spectrometry. The work makes use of most possibilities that SP HR-AMS gives and the whole text is clearly written using good English. Although the topic is important, there are several major issues that should be answered before publishing the paper.

First, although the use of collection efficiency 0.5 was common in the past, nowadays a composition dependent collection efficiency (Middlebrook et al., 2012) should be

used or at least tested as a composition is highly variable during sampling, as seen on diurnal variability of major species. This is even more important when no reference data for comparison with AMS total mass or any species are present.

Second, as automatic using of CE equal 0.5 induces higher uncertainty for determination of PM1 mass, it should be stated when it is compared to PM2.5 mass. The average ratio PM1 to PM2.5 equal to 0.54 is rather low especially when relatively low (for China) average concentrations were present during sampling period and at the same time average RH was not high.

Next, although using oxidation ratio of sulphur probably makes sense, the similar ratio for nitrogen (line 266) has no meaning as gas phase nitric acid is not included and there are also other nitrogen oxides than NO2 that are not accounted for.

The explanations about mainly photochemical origin of sulphate (lines 271-275) is misleading, the ratio SO42-/NO3- has nothing to do with sulphate origin. At the same time, although the maximum in oxidation ratio of sulphur in the afternoon may suggest influence of photochemical oxidation of SO2, it does not prove it. The same effect can be expected during increased mixed boundary layer period from down mixing of older aerosol from upper boundary layer in which the most of SO2 was oxidized some time ago. The size distribution of sulphate presented in Fig. 5 actually supports liquid phase formation of sulphates (Hering and Friedlander 1982).

The source apportionment as presented here is not very clear. The decision why the authors use four and not five factors seems little subjective. At the same time high correlation of "SV-OOA" factor with rBC, CO and NO2 (very similar to HOA factor) together with its very local origin (Fig. 11) does not seem to agree with its secondary origin. The doubts about SV-OOA in this work are also confirmed in the text and Table 1 because of its low correlation with SV-OOA from two other cities in China. The reviewer suggests testing five factor solution more thoroughly.

The text in lines 457-460 at least partially contradicts the conclusions based on Van

Krevelen diagram.

More attention should be paid to the differences between dual vaporizer and a tungsten vaporizer data, as it can bring more light on the influence of refractory organics.

The corrections and doubts described above should be also corrected in the Abstract (e.g. lines 54-55) and the Conclusions (lines 545, 553-554, 561-574).

---

## Author Comment (AC1) · 27 Jun 2016

**Response to Reviewer's Comments**

**Manuscript Number**:  acp-2016-226
**Authors:** Junfeng Wang, Xinlei Ge, Yanfang Chen, Yafei Shen, Qi Zhang, Yele Sun, Jianzhong Xu, Shun, Ge, Huan Yu, Mindong Chen

**Response to Reviewer #1 (Prof. Chen)**

**General comment:** This manuscript reports the measurement results of submicron aerosols by the SPAMS in Nanjing. Recently the Aerodyne AMS has been widely used around the world, and this work presents for the first time the results using the SP-AMS in the YRD region. This is overall a very well written paper with quite thorough analyses of the data, the figures are informative and the results provide new insights regarding the aerosol chemistry in this region.
Authors' reply: We thank Prof. Chen for his positive comment.

**Other comments:**

As the paper is submitted to the PEEX special issue, it will be good for the authors to describe the link of the results presented in this work with the overall scientific
goal of this special issue.
Authors' reply: As per the request, we have now added a sentence that describes the relationship of this work with PEEX, in the end of the introduction part, " *The findings for such a megacity are also valuable to the Pan-Eurasian Experiment (PEEX) infrastructure which aims to resolve the major uncertainties in Earth system science and global sustainability issues (Kulmala et al., 2015).*"

 "A constant collection efficiency (CE) of 0.5 was used for the mass quantification, in consistent with many other AMS studies, as indeed the mass fraction of ammonium nitrate (mostly <40%), particle acidity (near neutral) and RH (<10%) do not affect the CE significantly for this dataset (Middlebrook et al., 2012)." It is a bit difficult for non-AMS users to understand this point. Consider rephrasing to make the statement clear.
Authors' reply: As requested by the reviewer and also the CE issue raised by referee #3, we have now elaborated this issue in the revised manuscript.  "*A collection efficiency is typically used to account for the particles that are not measured by the instrument, due to the particles lost during passage through inlet, time-of-flight chamber and bouncing from the vaporizer.  For the SP-AMS, the CE of laser vaporizer is mainly governed by particle divergence, while for the tungsten vaporizer, the CE is governed mainly by the bouncing effects (Matthew et al., 2008). An AMS CE*

*value of 0.5 is typically valid and used commonly for most environments (Canagaratna et al., 2007). Nevertheless, Middlebrook et al. (2012) further found that high aerosol acidity, high ammonium nitrate mass fraction, and high sampling line RH could increase the CE, and provide a composition-dependent CE parameterization. For our dataset, we found that the composition-dependent CE rather than a constant CE=0.5 has negligible effects on the quantification of aerosol species, as the particles were neutralized (Fig. 3a), the mass fraction of ammonium nitrate were <40% in almost all cases, and also the sampling line RH was below 10%. In fact, the PM₁ mass concentrations by using the composition-dependent CE correlate a bit worse with the PM₂.₅ concentrations than ones using CE=0.5. For these reasons, we chose the constant CE of 0.5, in consistent with many other AMS studies, for this dataset.*"

In Fig.2, the authors used the $K_{p,AN}$ values to elucidate the formation mechanism of nitrate. Do the authors have gaseous $NH_3$ and $HNO_3$ data to make the argument more robust?

Authors' reply: Thanks for the suggestion. Unfortunately, we didn't have the co-located measurements of gaseous $NH_3$ and $HNO_3$, yet the theoretically estimated $K_{p,AN}$ values are likely adequate for illustrating the main formation pathway of nitrate.

In Fig.2b and Fig.10d, the authors separated the data into bins with 5ug/m3 increment, i think it will be better to make similar plots that describes the variations of aerosol compositions continuously against the mass loadings.

Authors' reply: As suggested, we have tried to re-plot the figure using all data instead of binned data. The new figures are shown below. However, due to the large amount and dramatic variations of the data, it is not easy to clearly observe the trends of particle compositions with the increase of PM₁ mass loadings; we thus kept the original plots in the new manuscript, as they seem to better help interpretation of the data.

[Figure]

It is interesting to use polar plots to demonstrate the characteristics of organic aerosols. How about similar plots for the inorganic species?

Authors' reply: The aim of the polar plots is to qualitatively describe the spatial distribution of the different aerosol species. As suggested, we have added the polar plots for nitrate, sulfate and total OA in Fig. 11. As can be seen from the new Fig.11. SV-OOA has overall similar distribution with nitrate, again verifying its semi-volatile behavior, while both sulfate and LV-OOA have broader distributions, suggesting their features as regional species. Relevant text added in the revised manuscript, "*Nitrate, as a semi-volatile species, behaves overall similar to the SV-OOA. High concentrations of LV-OOA are distributed in all directions under higher WS, similar to that of sulfate, representing their regional behaviors.*"

[Figure]

As the SP-AMS can measure both non-refractory and refractory species, it may provide unique information regarding the composition of refractory organics which cannot be measured by other AMS, this reviewers feel relevant discussion is lacking in the current manuscript.

Authors' reply: The reviewer perhaps didn't pay attention to Fig. 7, which compares the mass spectra obtained under dual-vaporizer setting and tungsten-only setting. It should be noted that, although under laser vaporization, some refractory organics can be detected, yet the SP-AMS cannot measure all refractory organics of the fine aerosols, but only the portion that is coated on *r*BC core. Overall, this portion seems to be negligible for this dataset, as can be seen in Fig. 7. We have also elaborate the discussion of this issue in the revised manuscript.

In Fig.12, the authors show the O/C vs. RH. Instead, i suggest to show the SV-OOA and LV-OOA mass concentrations with the increase of RH, so as to better demonstrate the influences of RH on the SOA formation.

Authors' reply: Thanks for the suggestion. We have added the plots which illustrate the SVOOA and LVOOA mass concentrations versus RH in the new manuscript. As shown below, both SVOOA and LVOOA mass concentrations don't increase with the increase of RH, the mass loading even start to decrease at high RH ranges, similar as the conclusions drawn from the O/C vs. RH plot, the results indicates insignificant influences from aqueous-phase oxidation for the SOA formation. Relevant text added in the revised manuscript, "*In addition, SV-OOA and LV-OOA mass concentrations, and O/C ratios of OA all show no obvious correlations with the RH as shown in Fig. 12b and Fig. 12c, indicating that aqueous-phase processing is insignificant compared to the photochemical processing for the oxidation of OA.*"

[Figure]

**General Comment**: The paper of Wang et al. describes the chemical characteristics of urban aerosol in Yangtze River Delta, China, measured by using a soot particle aerosol mass spectrometer. The results showed that most of the submicron particles consisted of organics, sulfate, nitrate, ammonium and black carbon. Authors used the results on chemical species to construct the light extinction, chemically resolved mass size distributions and the source apportionment of organics. Four factors were found for organic aerosol in Yangtze River Delta, hydrocarbon-like, cooking-related, semivolatile oxygenated and low-volatility oxygenated OA. Secondary OA dominated total organics, but when large OA concentrations were observed, the contribution of primary organics increased indicating the importance of anthropogenic sources.

This paper is well-written and the data have been analyzed and discussed very thoroughly. However, as there are so many AMS papers published on urban aerosols in past 15 years, it is difficult to see what is the significance and novelty of this paper. Are there any results presented in this paper that have not been published before? I suggest authors to think carefully what is the contribution of this paper to aerosol science and emphasize that clearly in the manuscript. Maybe it could be the unique features of the SP-AMS that allows to investigate the refractory material in addition to non-refractory species. In that case I suggest to focus this paper on that topic more clearly. This paper should be published after minor revision.

Authors' reply: We thank the referee for his/her overall positive comment on my work. Indeed, as a powerful tool for aerosol measurements, the AMS-related publications increased significantly in recent years. However, we believe that our results presented here are valuable and novel to the aerosol science: 1) Nanjing is one of the megacities in the densely populated Yangtze River delta region, but only two highly-time resolved studies using ACSM had been conducted during summer and winter for the aerosol characterization in Nanjing. Our study for the first time conducted the AMS measurement during springtime; 2) The ACSM cannot provide chemically-resolved size distribution of aerosol species, and cannot provide the HRMS of organic aerosols, while the high chemical resolution is sometimes critical for chemical speciation, and accurate source apportionment of OA; 3) ACSM cannot provide simultaneous measurement of BC. Overall, in this paper, we presented new data and new interpretation regarding the aerosol behaviors, sources and formation processes in Nanjing. These results are themselves never reported and insightful, enabling us to identify the diurnal patterns of different aerosol species, different sources that contribute to the aerosol mass, as well as the dominant pathway for the formation of secondary inorganic and organic aerosols, which are valuable for the abatement of

atmospheric pollution in China as well as in other megacities of the world. Relevant text was added in the introduction section, *"The rich highly-time resolved, highly-chemical resolved mass spectral data, as well as chemically-resolved size distributions of different aerosol species obtained for the first time in Nanjing during this study, can allow us to conduct in-depth analyses, and better understand the characteristics, sources and relevant transformation processes of ambient aerosols in Nanjing. The findings for such a megacity are also valuable to the Pan-Eurasian Experiment (PEEX) infrastructure which aims to resolve the major uncertainties in Earth system science and global sustainability issues (Kulmala et al., 2015)."*

Indeed, it is a unique feature that the SP-AMS is able to measure some refractory organics that other types of AMS cannot measure. We have added relevant discussion in Section 3.2, *"It should be noted that, accurate determination of refractory organics is very difficult, because: 1) A large portion of refractory organics cannot be detected by the SP-AMS if they didn't coat on rBC cores; 2) To accurately measure the species only coated on rBC cores, the tungsten vaporizer has to be physically removed, otherwise the vaporizer temperature is still around 150oC even its power is turned off, and the non-refractory organics that don't coat on rBC cores can still be measured, and complicates the analyses; 3) The CE and IE values for different species may vary under different vaporizer settings, so that direct subtraction of organics measured under tungsten-only setting from the organics measured under dual-vaporizer setting may not represent the real refractory organics; 4) Some ions measured under dual-vaporizer setting are likely induced by the laser itself rather than the 70 ev electron impact. For example, a series of fullerene-like carbon clusters can be generated by the laser itself, even though they don't really exist in the atmosphere (Wang et al., 2016;Onasch et al., 2015). This laser-induced ion formation scheme may work for other organics, thus makes it even more difficult for identifying the refractory organics. Further studies are essential to investigate this issue."* Moreover, we think this issue might be further explored by using a laser-only SP-AMS in parallel with a HR-AMS, and other instruments such as SP2, etc., for the specifically-designed chamber simulations that using *r*BC as seed aerosols, or for the real atmospheric environment where the *r*BC is heavily coated and internally mixed with other species. For this campaign, since we only have one SP-AMS without other supporting data, we are unable to conduct such in-depth investigations; in fact, we expect to elaborate this issue in more details in another publication as we had conducted such a campaign having HR-AMS, laser-only SP-AMS and other *r*BC instruments together, in Tibet.

**Specific comments:**
1. Page 2, line 34: remove (_54% of the PM2.5 mass) unimportant detail.

Authors' reply: Done

2. Page 5, line 124: Why springtime? Is there something specific in aerosol chemistry in springtime Yangtze River Delta? Add a reason for springtime measurements.

Authors' reply: During summertime, there may be influences from biomass burning and aqueous-phase oxidation, and during wintertime, there may be heating-related emissions in Nanjing. During springtime, the aerosol sources are likely different, yet there are no springtime AMS studies conducted before to probe the aerosol characteristics. Indeed, in this study, we found no influences from biomass burning, and coal-burning related factors, showing the different behaviors of springtime aerosols from those in other season. This has been stated clear in the introduction part of the revised manuscript now. *"Moreover, none of the previous AMS measurements studied the aerosol characteristics during springtime in Nanjing, yet the springtime aerosols may have different behaviors than those in other seasons, when aerosols are likely influenced significantly by emissions from biomass burning, coal burning etc."* In addition, we chose to conduct the measurement in the specific urban site is to help resolve the sources of aerosols in that site, as previous monitoring data shows a bit higher $PM_{2.5}$ level than those from adjacent sites. Thus, the findings in this paper also serve to the policy making of the local environmental protection agency.

3. Page 6, line 148: "coated species" Specify. What are the core and what are the coating species?

Authors' reply: Here, the "coated species" refer to the inorganic and organic components that coated on the $r$BC cores. It has been added in the revised manuscript.

4. Page 7: lines 183-186: collection efficiency, how is CE defined for the SP-AMS? Discuss with relevant citation.

Authors' reply: The CE issue is now elaborated in the revised manuscript. Please refer to the reply to reviewer #1.

5. Page 8, 206-210: no difference in OA factors between dual vaporizer and tungsten vaporizer; how about rBC? With dual vaporizer set-up you are able to separate rBC for different PMF factors. It would have been interesting to see how rBC divides between HOA and COA, or does it?

Authors' reply: We thank the referee for pointing out this question as we didn't state it clearly. Here, in this paper, we focused on the sources or non-refractory organics, thus we conducted PMF on the organics obtained under tungsten-only setting. We have also conducted PMF analyses on the organics obtained under dual-vaporizer setting but without including the $r$BC. As

shown in Fig.7, the organics mass spectra (with no *r*BC) are quite similar for these two circumstances, and we indeed found no significant difference from the PMF factors.

6. Page 17 and Fig 9; I suggest rather showing the correlation (time-series) of PMF factors and inorganic species than PMF factors and organic tracers in Figure 9.
Authors' reply: We now added the time series of nitrate and sulfate in Fig. 9.

7. Page 35, Figure 5a: Is it possible that the right side of the mass size distributions is defined by the transmission of aerodynamic lens? Could you estimate how much of the accumulation mode mass is missing for organics, sulfate, nitrate, ammonium and chloride because of that?

Authors' reply: The reviewer is correct. AMS has different transmission efficiencies for particles with different sizes due to the inlet system. Fluid dynamic simulation of the AMS inlet shows that the AMS shows 100% transmission efficiency for 70-500nm particles, and substantial transmission for small particles (30-70nm) and large particles (500nm-2.5μm) for spherical particles (Jayne et al., 2000). The AMS is referred to as a $PM_1$ instrument, as its transmission efficiency at 1μm is approximately 50%. Recently, there is a new lens system that can efficiently transmit supermicron particles up to 3 μm (Williams et al., 2013), but instead its transmission efficiency for small particles (left side) was significantly decreased.
Overall it is very difficult to estimate how much of the mass is missing due to the incomplete transmission of our SP-AMS inlet system. A possible way is to inject the DMA-selected monodisperse single-component particles (pure ammonium nitrate, ammonium sulfate, ammonium chloride, etc), and compares the AMS-measured numbers with the particle numbers counted by the CPC – however, this method is also limited by the upper size cut of the DMA (typically ~700 nm), and also there is no proper reference material for the estimation of organics. Nevertheless, the AMS is able to capture the bulk of ambient accumulation mode particles in the submicron meter range, relevant discussion and analyses mainly focus on the peak modes in $PM_1$ range as well. We have added a sentence to elaborate this point in section 3.2, "*It also should be note that, although the AMS is able to capture the bulk of atmospheric accumulation mode particles (Canagaratna et al., 2007), right side of size distributions may be affected by the incomplete transmission of larger particles limited by the SP-AMS inlet (in particular, the supermicron ones).*"

Technical corrections:
Page 3, lines 67-68: parenthesis are used unclearly

Page 15, line 428: …the SV-OOA… remove "the" as you haven't used it with PMF factors earlier

Authors' reply: Corrected.

Response to Anonymous Referee #3

**General comments:** The paper deals with highly time-resolved urban aerosol characterization during two weeks period in spring in Nanjing (China) using soot particle high resolution aerosol mass spectrometry. The work makes use of most possibilities that SP HR-AMS gives and the whole text is clearly written using good English. Although the topic is important, there are several major issues that should be answered before publishing the paper.

Authors' reply: We thank the referee for his/her overall positive comment on my work, and we have tired our best to address the specific comments below.

First, although the use of collection efficiency 0.5 was common in the past, nowadays a composition dependent collection efficiency (Middlebrook et al., 2012) should be used or at least tested as a composition is highly variable during sampling, as seen on diurnal variability of major species. This is even more important when no reference data for comparison with AMS total mass or any species are present. Second, as automatic using of CE equal 0.5 induces higher uncertainty for determination of PM1 mass, it should be stated when it is compared to PM2.5 mass. The average ratio PM1 to PM2.5 equal to 0.54 is rather low especially when relatively low (for China) average concentrations were present during sampling period and at the same time average RH was not high.

Authors' reply: As also requested by reviewer #1, we have now elaborated this issue in the revised manuscript. *"A collection efficiency is typically used to account for the particles that are not measured by the instrument, due to the particles lost during passage through inlet, time-of-flight chamber and bouncing from the vaporizer. For the SP-AMS, the CE of laser vaporizer is mainly governed by particle divergence, while for the tungsten vaporizer, the CE is governed by the bouncing effects (Matthew et al., 2008). An AMS CE value of 0.5 is typically valid and used commonly for most environments (Canagaratna et al., 2007). Nevertheless, Middlebrook et al. (2012) further found that high aerosol acidity, high ammonium nitrate mass fraction, and high sampling line RH could increase the CE, and provide a composition-dependent CE parameterization. For our dataset, we found that the composition-dependent CE rather than a constant CE=0.5 has negligible effects on the quantification of aerosol species, as the particles*

*were neutralized (Fig. 3a), the mass fraction of ammonium nitrate were <40% in almost all cases, and also the sampling line RH was below 10%. In fact, the $PM_1$ mass concentrations by using the composition-dependent CE correlate a bit worse with the $PM_{2.5}$ concentrations than ones using CE=0.5. For these reasons, we chose the constant CE of 0.5, in consistent with many other AMS studies, for this dataset."*

In addition, the measured ratio of 0.54 for $PM_1$ to $PM_{2.5}$ seems a bit low, but is close to 0.63 for $PM_1$ to $PM_{2.5}$ measured by the ACSM during Nanjing Winter by Zhang et al. (2016), and higher than 0.38 measured by the HR-AMS during Lanzhou Summer by Xu et al. (2014). We have added a sentence to state clearly the uncertainty due to CE of the SP-AMS. *"This ratio appears to be a bit low, likely due to the uncertainty of CE of the SP-AMS."*

Next, although using oxidation ratio of sulphur probably makes sense, the similar ratio for nitrogen (line 266) has no meaning as gas phase nitric acid is not included and there are also other nitrogen oxides than NO2 that are not accounted for. The explanations about mainly photochemical origin of sulphate (lines 271-275) is misleading, the ratio SO42-/NO3- has nothing to do with sulphate origin. At the same time, although the maximum in oxidation ratio of sulphur in the afternoon may suggest influence of photochemical oxidation of SO2, it does not prove it. The same effect can be expected during increased mixed boundary layer period from down mixing of older aerosol from upper boundary layer in which the most of SO2 was oxidized some time ago. The size distribution of sulphate presented in Fig. 5 actually supports liquid phase formation of sulphates (Hering and Friedlander 1982).

Authors' reply: Thanks for pointing out this issue. We agree with the reviewer that due to possible existence of other nitrogen oxides and gaseous nitric acid, the $f_N$ defined here may be not a good proxy to represent the oxidation of $NO_2$, so we have now removed Fig.4b, 4c, 4e and 4f. Instead, we added one plot that depicts the relationship of nitrate and sulfate concentrations with RH. From Fig. 4b and 4c, we can find that the $f_s$ shows anti-correlation with the RH, and also, the mass concentrations of sulfate don't show clearly positive correlation with the RH – the mass loadings even start to drop under high RH conditions. We postulate that at least it suggests that aqueous-phase oxidation doesn't show significant influence on sulfate formation. Indeed, we cannot completely exclude the possibility that increase of $f_s$ or sulfate during afternoon might be due to down mixing of sulfate that formed some time ago. However, on the other hand, as all aerosol species are mixed together, the down mixing effect may have also increased the concentrations of other species as well - which was indeed observed before in Lanzhou summer (Xu et al., 2014). Yet concentrations of all other species, on the contrary, decreased significantly

during afternoon. We thus think these behaviors suggest that the afternoon photochemical formation of sulfate is likely dominant rather than the aqueous-phase processing. The size distributions data may have limited use here for describing the formation pathway of sulfate or nitrate - as all secondary inorganic species peak in a similar mode size around 500-700nm (the peak size of nitrate even appears to be a bit larger than that of sulfate).

[Figure]

The text regarding Fig. 4 is now re-written. "*In order to further elucidate the formation processes of sulfate, we calculated the oxidation ratios of sulfur ($f_S$) (Fig. 4a), defined as $f_S = nSO_4^{2-}/(nSO_4^{2-} + nSO_2)$ (Xu et al., 2014), indicating the conversion of $SO_2$ . Here $nSO_4^{2-}$ and $nSO_2$ are the molar quantities of particle-phase sulfate, and gas-phase $SO_2$, respectively. Diurnal variations of $f_S$ and RH are presented in Fig.4b, and Fig. 4c shows variations of sulfate and nitrate concentrations with RH. The diurnal profile of $f_S$ shows a negative correlation with that of RH (r = -0.52), and mass concentrations of sulfate even drop under high RH conditions, indicating somewhat an insignificant role of aqueous-phase process for sulfate formation during this campaign. On the other hand, the $f_S$ reaches a maximum around 3 pm. Note the afternoon rise of fs and sulfate may be affected by the down mixing of sulfate formed earlier, however, since concentrations of all other aerosol species that mix with sulfate decrease significantly, we postulate that the increase of fs likely suggest the photochemical production of sulfate in the afternoon.*"

The source apportionment as presented here is not very clear. The decision why the authors use four and not five factors seems little subjective. At the same time high correlation of "SV-OOA"

factor with rBC, CO and NO2 (very similar to HOA factor) together with its very local origin (Fig. 11) does not seem to agree with its secondary origin. The doubts about SV-OOA in this work are also confirmed in the text and Table 1 because of its low correlation with SV-OOA from two other cities in China. The reviewer suggests testing five factor solution more thoroughly.

Authors' reply:  Sorry that we didn't put enough details regarding the choice of PMF factors. In fact, we have investigated the 5-factor solution space carefully before choosing 4-factor one as the best solution. 1) 4-factor SVOOA mass spectrum does correlate a bit worse with the SVOOA from Beijing and Lanzhou, compared with the correlation coefficients of other factors. But it does correlate very well with the OOA from Fresno. Further investigation finds that the relative weak correlation ($r^2$= 0.75) between 4-factor SVOOA with Beijing SVOOA is mainly due to 3 ions, $CO_2^+$, $CO^+$ and $H_2O^+$ (in theory, only $CO_2^+$, as the other two ions are scaled to $CO_2^+$); and if we exclude these 3 ions, $r^2$ will be 0.88. Similarly, the weaker correlation with Lanzhou SVOOA is mainly due to $C_2H_3O^+$, if we excluded this ion, $r^2$ will be 0.89. These results suggest that the 4-factor SVOOA MS is in consistent with previous studies.

2)Correspondingly, although correlations between factor 2 of 5-factor solution ("SVOOA" in 5-factor solution) with Beijing and Lanzhou SVOOA does improve a little than "SVOOA" in 4-factor solution ($r^2$=0.72 vs. 0.68 with Beijing, and $r^2$=0.80 vs. 0.75 with Lanzhou), the correlation with Fresno OOA on the other hand, becomes a bit worse ($r^2$= 0.87 vs. 0.91). Thus we think there is no specific reason to conclude that the SVOOA in 5-factor solution is better than the SVOOA in 4-factor solution, at least based on the spectral similarity.

3) "SVOOA" in 5-factor solution almost has no correlation with nitrate (r=0.07), while the SVOOA in 4-factor solution correlates relatively well with nitrate (r=0.49).

4) Factor 4 in 5-factor solution is hard to explain. It is not BBOA because of negligible signals at m/z 60 and 73. Its spectrum is not similar to HOA ($r^2$=0.33, 0.24 and 0.26 with HOA of Beijing, Lanzhou and Fresno), COA ($r^2$= 0.27, 0.39 and 0.40 with COA of Beijing, Lanzhou and Fresno), or SVOOA ($r^2$=0.42 and 0.48 with SVOOA of Beijing and Lanzhou).

5) Factor 4 and Factor 2 in the 5-factor solution seems to be a split of SVOOA in 4-factor solution. First, the O/C of 0.32 of SVOOA in 4-factor solution is between 0.28 of Factor 4 and 0.42 of Factor 2 in the 5-factor solution; more importantly, if we combined Factor 2 and Factor 4 into one "Combined factor", its time series correlate very well with the SVOOA in 4-factor solution ($r^2$=0.89) with a slope of 1.05, suggesting clearly the factor splitting.

(6)Our results do suggest that SVOOA is locally formed, it has good correlations with rBC, and HOA. We think this does not necessarily mean that the PMF is not correct, or the SVOOA we separate is in fact a primary OA factor. More likely, it is a locally formed fresh SOA, which

derived from the in-situ fast oxidation of traffic or cooking VOCs. Of course, this issue requires more careful investigation.

[Figure]

We have added some discussions into the revised manuscript. "*Following the instruction detailed by Zhang et al. (2011), the 4-factor solution (at fpeak = -0.1) was chosen as the optimal solution, as the 3-factor solution cannot separate the hydrocarbon-like OA (HOA) and cooking OA (COA) (Fig. S2). For the 5-factor solution (Fig. S3), Factor 2 and Factor 4 are clearly a split from the SVOOA from the 4-factor solution (r2 = 0.89 and slope of 1.05); Factor 2 of 5-factor solution also shows much weaker correlations with nitrate than SVOOA of 4-factor solution does (r = 0.07 vs. 0.49).*"

The text in lines 457-460 at least partially contradicts the conclusions based on Van Krevelen diagram.
Authors' reply: Thanks for pointing out this issue. We have modified the relevant discussion for the Van Krevelen diagram. The VK diagram should be used with cautions as for ambient data it cannot exclude mixing effects. The data follows a line with a slope of -1 may not reflect the real aging mechanism is the addition of carboxylic acid. We thus delete the statement "the propagation of OA is similar to an aging process that is likely driven by the addition of carboxylic acid (slope of -1)".

More attention should be paid to the differences between dual vaporizer and a tungsten vaporizer data, as it can bring more light on the influence of refractory organics.
Authors' reply: We thank the referee for his/her suggestion. We have added some discussion regarding this issue. Please check the reply to reviewer #2.

The corrections and doubts described above should be also corrected in the Abstract
(e.g. lines 54-55) and the Conclusions (lines 545, 553-554, 561-574).
Authors' reply: We have modified accordingly the Abstract and Conclusions in the revised manuscript.

References

Canagaratna, M. R., Jayne, J. T., Jimenez, J. L., Allan, J. D., Alfarra, M. R., Zhang, Q., Onasch, T. B., Drewnick, F., Coe, H., Middlebrook, A., Delia, A., Williams, L. R., Trimborn, A. M., Northway, M. J., DeCarlo, P. F., Kolb, C. E., Davidovits, P., and Worsnop, D. R.: Chemical and microphysical characterization of ambient aerosols with the aerodyne aerosol mass spectrometer, Mass Spectrom. Rev., 26, 185-222, 10.1002/Mas.20115, 2007.

Jayne, J. T., Leard, D. C., Zhang, X., Davidovits, P., Smith, K. A., Kolb, C. E., and Worsnop, D. R.: Development of an Aerosol Mass Spectrometer for Size and Composition Analysis of Submicron Particles, Aerosol Sci. Tech., 33, 49 - 70, 10.1016/S0021-8502(98)00158-X, 2000.

Kulmala, M., Lappalainen, H. K., Petäjä, T., Kurten, T., Kerminen, V. M., Viisanen, Y., Hari, P., Sorvari, S., Bäck, J., Bondur, V., Kasimov, N., Kotlyakov, V., Matvienko, G., Baklanov, A., Guo, H. D., Ding, A., Hansson, H. C., and Zilitinkevich, S.: Introduction: The Pan-Eurasian Experiment (PEEX) – multidisciplinary, multiscale and multicomponent research and capacity-building initiative, Atmos. Chem. Phys., 15, 13085-13096, 10.5194/acp-15-13085-2015, 2015.

Matthew, B. M., Middlebrook, A. M., and Onasch, T. B.: Collection Efficiencies in an Aerodyne Aerosol Mass Spectrometer as a Function of Particle Phase for Laboratory Generated Aerosols, Aerosol Sci Tech, 42, 884-898, 10.1080/02786820802356797, 2008.

Middlebrook, A. M., Bahreini, R., Jimenez, J. L., and Canagaratna, M. R.: Evaluation of Composition-Dependent Collection Efficiencies for the Aerodyne Aerosol Mass Spectrometer using Field Data, Aerosol Sci. Tech., 46, 258-271, 10.1080/02786826.2011.620041, 2012.

Onasch, T. B., Fortner, E. C., Trimborn, A. M., Lambe, A. T., Tiwari, A. J., Marr, L. C., Corbin, J. C., Mensah, A. A., Williams, L. R., Davidovits, P., and Worsnop, D. R.: Investigations of SP-AMS Carbon Ion Distributions as a Function of Refractory Black Carbon Particle Type, Aerosol Sci. Tech. , 49, 409-422, 10.1080/02786826.2015.1039959, 2015.

Wang, J., Onasch, T. B., Ge, X., Collier, S., Zhang, Q., Sun, Y., Yu, H., Chen, M., Prévôt, A. S. H., and Worsnop, D. R.: Observation of Fullerene Soot in Eastern China, Environmental Science & Technology Letters, 3, 121-126, 10.1021/acs.estlett.6b00044, 2016.

Williams, L. R., Gonzalez, L. A., Peck, J., Trimborn, D., McInnis, J., Farrar, M. R., Moore, K. D., Jayne, J. T., Robinson, W. A., Lewis, D. K., Onasch, T. B., Canagaratna, M. R., Trimborn, A., Timko, M. T., Magoon, G., Deng, R., Tang, D., de la Rosa Blanco, E., Prévôt, A. S. H., Smith, K. A., and Worsnop, D. R.: Characterization of an aerodynamic lens for transmitting particles greater than 1 micrometer in diameter into the Aerodyne aerosol mass spectrometer, Atmos. Meas. Tech., 6, 3271-3280, 10.5194/amt-6-3271-2013, 2013.

Xu, J., Zhang, Q., Chen, M., Ge, X., Ren, J., and Qin, D.: Chemical composition, sources, and processes of urban aerosols during summertime in northwest China: insights from high-resolution aerosol mass spectrometry, Atmos. Chem. Phys., 14, 12593-12611, 10.5194/acp-14-12593-2014, 2014.

Zhang, Q., Jimenez, J., Canagaratna, M., Ulbrich, I., Ng, N., Worsnop, D., and Sun, Y.: Understanding atmospheric organic aerosols via factor analysis of aerosol mass spectrometry: a review, Anal. Bioanal. Chem., 401, 3045-3067, 10.1007/s00216-011-5355-y, 2011.

Zhang, Y. J., Tang, L., Yu, H., Wang, Z., Sun, Y., Qin, W., Chen, W., Chen, C., Ding, A., Wu, J., Ge, S., Chen, C., and Zhou, H.-c.: Chemical composition, sources and evolution processes of aerosol at an urban site in Yangtze River Delta, China during wintertime, Atmos. Environ., 123, 339-349, 10.1016/j.atmosenv.2015.08.017, 2016.